# Topological deep learning for enhancing peptide-protein complex prediction
Xuhang Dai[1,4], Rui Wang [2,4] & Yingkai Zhang [1,2,3] ✉

Peptide-protein interactions are essential to biological processes and drug discovery, but selecting high-quality models from predicted complexes remains challenging due to high false positive rates (FPR). Here we introduce TopoDockQ, a topological deep learning model leveraging persistent combinatorial Laplacian (PCL) features to predict DockQ scores (p-DockQ) for accurately evaluating peptide-protein interface quality, aimed at enhancing precision and mitigating FPR in model selection. Compared to AlphaFold2's built-in confidence score, TopoDockQ reduces false positives by at least 42% and increases precision by 6.7% across five evaluation datasets filtered to ≤70% peptide-protein sequence identity, while maintaining relatively high recall and F1 scores. To support flexible peptide design, we introduce ResidueX, a workflow incorporating non-canonical amino acids (ncAA) into peptide scaffolds. Together, TopoDockQ and ResidueX advance peptide-protein modeling by refining confidence scoring and supporting ncAA incorporation, enabling precise, customizable design and accelerating next-generation peptide therapeutics development.

Peptide-protein interactions are pivotal in numerous biological phenomena, encompassing signal transduction[1–3], enzyme cleavage[4], and immune response[5]. The precision of peptide-protein binding mode and affinity[6,7] is crucial for both comprehending molecular mechanisms and rational design of therapeutic peptides. Methods such as X-ray crystallography[8], nuclear magnetic resonance spectroscopy[9], and cryo-electron microscopy[10] are typically employed for the structural characterization of peptide-protein complexes. However, these techniques can be laborious, costly, and not universally applicable across all systems. Consequently, computational methods, particularly molecular docking[11], have emerged as complementary to experimental approaches for predicting peptide-protein interactions, offering rapid and cost-effective insights into the structural and energetic aspects of binding. Conventional peptide docking methods can be categorized into three types: template-based docking, local docking, and global docking. Template-based docking, exemplified by GalaxyPepDock[12], performs similarity-based docking based on experimentally determined structures in the PepBind database[13] and energy-based optimization. Local docking employs small molecule docking algorithms, such as AutoDock Vina[14], GOLD[15], and Surflex-Dock[16], as well as specific algorithms like AutoDock CrankPep[17] and HADDOCK peptide docking[18].

Evaluation of global docking methods designed for situations where peptide binding sites are unknown highlights the challenges inherent in this task. Among these traditional docking methods, PIPER-FlexPepDock[19] demonstrates superior performance compared to other global docking algorithms such as pepATTRACT[20] and HADDOCK[18,19]. Nevertheless, their overall success rates remain below 50% and are generally lower than those achieved by local docking methods or approaches guided by known binding motifs. In cases where the binding site is not well-defined or experimentally validated, global docking provides a feasible approach for modeling peptide-protein interactions. It is also well-suited for investigating secondary or alternative binding sites beyond those identified a priori. The relatively low success rates in peptide docking are likely due to two major challenges: the significant conformational changes[11,21–24] in both protein and peptide during binding, attributed to their high backbone flexibility, and the difficulties in accurately scoring and ranking docked poses[11,24]. Drawing inspiration from the similarity between the peptide-protein binding process and protein folding, MDockPeP2[25] has been developed to address these challenges by utilizing physicochemical information inherent in numerous monomeric proteins, combined with an exhaustive search method and an integrated approach that includes global search and local flexible minimization techniques. However, MDockPeP2 still requires a protein structure as input, and in their study, the apo form was selected to enable site-independent global peptide-protein docking without prior knowledge of the binding site.

With the advent of artificial intelligence, applications have emerged in various fields, from small molecule docking to biomolecular structure prediction. AlphaFold2 (AF2), with its groundbreaking deep learning approach, has demonstrated substantial improvements in accuracy in

[1]Department of Chemistry, New York University, New York, NY, USA. [2]Simons Center for Computational Physical Chemistry, New York University, New York, NY, USA. [3]NYU-ECNU Center for Computational Chemistry, New York University Shanghai, Shanghai, CN, China. [4]These authors contributed equally: Xuhang Dai, Rui Wang. ✉e-mail: yingkai.zhang@nyu.edu

predicting protein-protein complex structures, making it a promising tool for enhancing peptide-protein docking predictions. Tsaban et al.[23] and Shanker et al.[26] have tried these deep learning techniques such as OmegaFold[27], AlphaFold2-Monomer[28], and AlphaFold2-Multimer (AF2-M)[29], in peptide-protein docking tasks, showing a general success rate higher than 50% with AF2-M. More recently, AlphaFold3 (AF3), with its incorporation of diffusion-based modeling, has further advanced the accuracy of protein structure predictions, providing opportunities for addressing complex docking scenarios. RFdiffusion[30] and its derivatives RFpeptides[31], are primarily designed for de novo protein and macrocyclic peptide design rather than docking specific peptide sequences, including those with non-canonical amino acids (ncAAs), to protein targets. For ncAA peptide-protein complex modeling, alternative approaches like the Rosetta Suite[32,33], also face challenges such as manual parameterization and limited force field compatibility for ncAAs.

Despite the remarkable advancements of AF2-M and AF3, there remain exciting opportunities to further expand their applicability in peptide docking tasks. In particular, two key challenges should be addressed to enhance peptide-protein complex prediction. First, a more accurate model selection is needed as the built-in confidence score (af_confidence) in AF2-M and AF3 often leads to a high incidence of false positives in practical applications. This limitation reduces the efficiency of peptide-protein complex prediction and increases computational overhead in downstream analyses. Second, enhanced support for ncAAs is essential to broaden the scope of deep learning-based structure prediction. ncAAs play a critical role in improving peptide stability, bioavailability, and specificity, making them essential for advanced therapeutic peptide design. However, current implementations of AF2-M and AF3 primarily support only natural amino acids, restricting their predictive utility to conventional peptides.

To address these challenges, we introduce two complementary approaches. First, we employ a well-established mathematical tool, persistent combinatorial Laplacian (PCL)[34–37], to extract substantial topological changes and shape evolution features from the peptide-protein interface. These features are utilized to develop TopoDockQ (see Fig. 1), a predictive topological deep learning (TDL) model capable of predicting the DockQ score. The DockQ score is a metric ranging from 0 to 1 that serves as a pivotal indicator of the quality of the peptide-protein interface, thereby facilitating the selection of the most suitable peptide-protein complex model. To mitigate data leakage and evaluate the generalization of TopoDockQ, we generated five filtered datasets, including LEADSPEP_70%, Latest_70%, ncAA-1_70%, PFPD_70%, and SinglePPD-Test_70%, each retaining only peptide-protein complexes with ≤ 70% sequence identity to the training data. By predicting DockQ scores (p-DockQ), TopoDockQ enables accurate and reliable model selection by achieving at least 42% reductions in false positive rate (FPR) and 6.7% increment in precision across all five evaluation datasets filtered at a 70% peptide-protein sequence identity threshold, while maintaining recall and F1 scores at relatively high levels compared to AF2's built-in confidence score. Second, we introduce the ResidueX workflow to enable the integration of ncAAs into peptide docking pipelines. This workflow builds on the natural peptide scaffolds generated by AF2-M and AF3 and prioritizes scaffolds based on their p-DockQ scores. By systematically introducing ncAAs into the peptide structures, the ResidueX workflow extends the capabilities of AF2-M and AF3 to include non-canonical peptides.

In summary, this study contributes to the advancement of peptide-protein complex prediction within the AlphaFold framework by enhancing the model selection step through TopoDockQ and enabling subsequent non-canonical amino acid incorporation via the ResidueX workflow. Together, these innovations provide a foundation for the rational design of therapeutic peptides, expanding the possibilities in peptide engineering and drug discovery.

## Results and discussion
In this study, we first evaluate the peptide-protein complex prediction performance of AF2-M and AF3, and then compare it with other existing methods. To address the issue of a high FPR associated with the AF2-M and AF3 built-in confidence score during model selection, we develop TopoDockQ, a TDL model based on persistent topological Laplacian-based features, to predict the DockQ score, which ranges from 0 to 1, aiming to mitigate the high FPR problem. Additionally, we establish ResidueX, a workflow for the generation of non-canonical peptides, to extend AF2-M and AF3's peptide-protein complex prediction capabilities by incorporating various ncAAs. The predicted DockQ score (p-DockQ) is then used to select high-quality peptide scaffolds and accurately model ncAA-containing peptide conformations, improving model selection reliability and broadening the applicability of structure prediction tools. To support the development and evaluation of our models, we used the SinglePPD dataset for training (SinglePPD-Training), validation (SinglePPD-Validation), and testing (SinglePPD-Testing) of TopoDockQ. Model performance is further assessed on five datasets: SinglePPD_Test, PFPD, LEADSPEP, Latest, and ncAA-1. Here, the SinglePPD dataset is derived from BioLip[38] and includes natural, linear protein-peptide complexes with no covalent bonds. It is divided into training (1837 samples), validation (193 samples), and test (191 samples) subsets. The Latest dataset contains recent depositions (from January 2023 to May 2024) of natural complexes with peptide lengths under 30 residues. PFPD and PFPD_42 are benchmark subsets with 27 non-redundant and 42 redundant samples, respectively. LEADSPEP includes 43 single protein-single peptide complexes curated from an established benchmark. Lastly, the ncAA-1 dataset contains 150 complexes featuring ncAAs. Further details on all datasets are provided in Table 1. To mitigate potential data leakage and evaluate model generalization, we derived five "70%" evaluation subsets: SinglePPD-Test_70%, PFPD_70%, LEADSPEP_70%, Latest_70%, and ncAA-1_70%, by retaining only complexes with a protein-peptide sequence identity product of ≤ 70% relative to the training set. As summarized in Table 2, the resulting subsets (mean values denote the average protein-peptide sequence identity product; ranges in parentheses) comprise LEADSPEP_70% (33 complexes; mean: 0.22, range: 0.00–0.67), Latest_70% (75 complexes; mean: 0.12, range: 0.00–0.60), ncAA-1_70% (138 complexes; mean: 0.16, range: 0.00–0.70), PFPD_70% (20 complexes; mean: 0.14, range: 0.00–0.57), and SinglePPD-Test_70% (118 complexes; mean: 0.24, range: 0.00–0.68), thereby satisfying the intended cutoff while preserving sufficient diversity for effective evaluation.

### AlphaFold2-multimer and AlphaFold3 are promising for peptide-protein complex prediction
By treating peptides as short proteins, peptide-protein complex prediction was performed using AF2-M and AF3. Both protein and peptide sequences were prepared in FASTA format, with 50 models generated per complex by AF2-M and 5 models per complex by AF3. For performance evaluation, we define a successful prediction as one in which at least one generated model had a peptide backbone RMSD ≤ 3.0 Å. To benchmark their performance against existing methods, we evaluate AF2-M and AF3 on the PFPD dataset, a non-redundant set of 27 peptide-protein complexes. AF3 and AF2-M can achieve success rates of 81% and 67%, respectively, outperforming traditional peptide-protein complex prediction methods (Supplementary Fig. 1A) such as PIPER-FlexPepDock, pepATTRACT and HADDOCK.

As shown in Supplementary Fig. 1A), we also evaluated NeuralPLexer[39], a recent diffusion-based generative model initially developed for protein-ligand complex prediction. Although being the latest diffusion model, NeuralPLexer failed to generalize to peptide-protein complex prediction tasks under default settings on the PFPD dataset. This limitation may stem from several factors, including the lack of MSA input, the high flexibility of peptide ligands, and the model's training focus on protein-ligand systems, resulting in fewer peptide-protein complexes in its training data. In addition, we note that, among all the evaluation datasets used in this study, only the PFPD dataset was mentioned in this section as it is widely adopted by other research groups, which facilitates direct comparisons.

Unlike traditional peptide docking algorithms that require a known protein structure, AF2-M and AF3 rely solely on sequence information. As a result, both the protein and peptide backbone RMSDs were evaluated. The

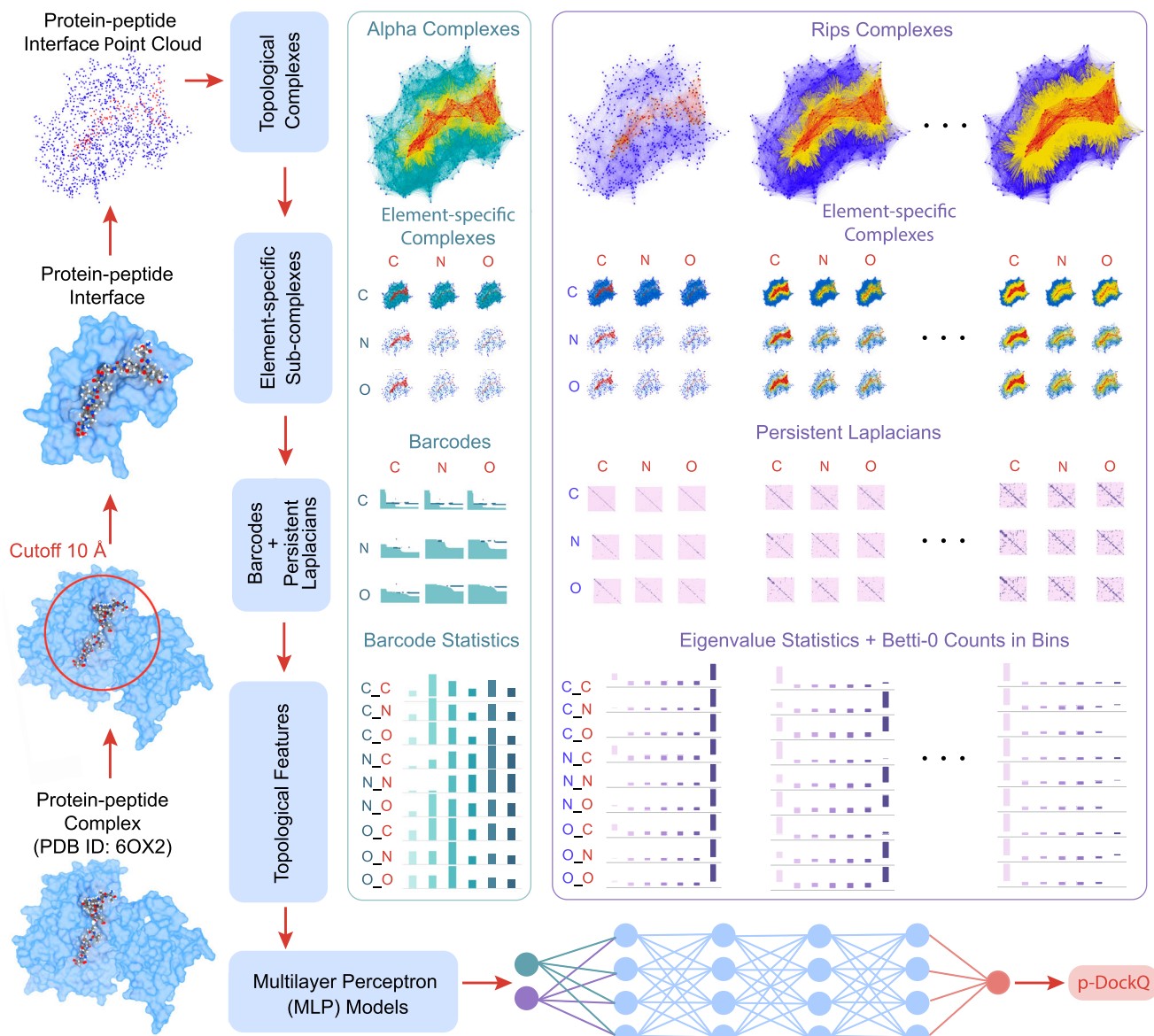

**Fig. 1 | The general workflow of TopoDockQ, a topological deep learning model for predicting DockQ values.** The process begins by defining the peptide-protein interface using a 10 Å cutoff, capturing the 3D coordinates of carbon (C), nitrogen (N), and oxygen (O) atoms as a point-cloud dataset. From this dataset, we construct Alpha and Rips complexes and generate nested families of element-specific sub-complexes along distance-based filtration parameters. Then, we extract features using persistent combinatorial Laplacian-based feature generation, yielding 378 barcode statistics features, 2079 eigenvalue statistics features, and 189 Betti-0 counts in bins features. These features are concatenated into a 1D feature array (dimension 2646), serving as the input to train a multilayer perceptron (MLP) model that can predict DockQ values. This MLP consists of four hidden layers with 2048 neurons each and is trained using batch normalization and Xavier initialization to ensure stability. Training is conducted on an NVIDIA A100 GPU with a batch size of 512 and a learning rate of 0.0005.

distribution of protein parts backbone RMSD (Supplementary Fig. 1B, C) is mostly within 2 Å. The peptide backbone RMSD values vary greatly in the range of 0 and higher than 10 Å. Additionally, AF2-M was chosen over AF3 for generating training data due to AF3's usage restrictions, which prohibit the use of its predicted structures for training other machine-learning models. Given this limitation, AF2-M remains the optimal choice for dataset generation in this study.

### Improved precision and reduced false positive rate with p-DockQ score

When evaluating the peptide-protein complexes generated by AF2-M on the PFPD dataset, the FPR was as high as 0.42 (Supplementary Fig. 2A) when selecting the top-1 model ranked by the AlphaFold built-in confidence score (af_confidence). The FPR is even higher when selecting the top-1

model from AF3 predictions (Supplementary Fig. 3A). This motivated the development of a more effective confidence score to improve model selection. Given that the DockQ score is a scaled metric ranging from 0 to 1, it serves as a more suitable target for regression by naturally constraining the output range and stabilizing training.

TDL models, particularly those leveraging PCL features, are well-established in capturing structural and relational patterns in biomolecular systems. PCL-based features offer key advantages by encoding multi-scale geometric and topological properties of molecular interactions, providing a more comprehensive representation of peptide-protein interfaces than traditional structural descriptors.

Therefore, we develop TopoDockQ, a TDL framework that predicts DockQ scores directly from the 3D structure of peptide-protein interfaces. The interface is represented as a point-cloud, from which we extract

**Table 1 | Overview of datasets used for training, validation, and evaluation of peptide-protein interaction models**

| Dataset name | Size | Dataset type | Natural peptide | Description |
|---|---|---|---|---|
| SinglePPD-Training | 1837 | Training | Yes | Derived from BioLip[38]. Each complex contains one protein and one natural, linear peptide, with no covalent bond between the protein and the peptide. The dataset includes structures released up to January 1, 2021. |
| SinglePPD-Validation | 193 | Validation | Yes | Same data source and filtering criteria as SinglePPD-Training. |
| SinglePPD-Test | 191 | Evaluation | Yes | Same data source and filtering criteria as SinglePPD-Training. |
| SinglePPD-Test_70% | 118 | Evaluation | Yes | Derived from SinglePPD-Test dataset. Peptide-protein complexes with low sequence similarity to the training data, defined by a similarity product ≤ 70%, ensuring that at least one component, either the protein or the peptide, has low similarity to the training set. |
| SinglePPD-Test_LowSimilarity | 80 | Evaluation | Yes | Peptide-protein complexes with low sequence similarity to the training data, defined by a similarity product ≤ 30%, ensuring that at least one component, either the protein or the peptide, has low similarity to the training set. |
| Latest | 89 | Evaluation | Yes | Natural, linear peptides (residue length ≤ 30) with a deposition date after January 1, 2023; protein structures were collected between January 1, 2023, and May 1, 2024. |
| Latest_70% | 75 | Ealuation | Yes | Derived from Latest dataset. Peptide-protein complexes with low sequence similarity to the training data, defined by a similarity product ≤ 70%, ensuring that at least one component, either the protein or the peptide, has low similarity to the training set. |
| PFPD | 27 | Evaluation | Yes | Non-redundant peptide-protein complexes used in the study of PIPER-FlexPepDock[19]. |
| PFPD_70% | 20 | Evaluation | Yes | Derived from PFPD dataset. Peptide-protein complexes with low sequence similarity to the training data, defined by a similarity product ≤ 70%, ensuring that at least one component, either the protein or the peptide, has low similarity to the training set. |
| PFPD_42 | 42 | Evaluation | Yes | Redundant peptide-protein complexes used in the study of PIPER-FlexPepDock. |
| LEADSPEP | 43 | Evaluation | Yes | 43 out of 53 LEADS-PEP[47] entries containing only single protein-single peptide complexes. |
| LEADSPEP_70% | 33 | Evaluation | Yes | Derived from LEADSPEP dataset. Peptide-protein complexes with low sequence similarity to the training data, defined by a similarity product ≤ 70%, ensuring that at least one component, either the protein or the peptide, has low similarity to the training set. |
| ncAA-1 | 150 | Evaluation | No | Linear peptides with exactly one non-canonical amino acid. |
| ncAA-1_70% | 138 | Evaluation | No | Derived from ncAA-1 dataset. Peptide-protein complexes with low sequence similarity to the training data, defined by a similarity product ≤ 70%, ensuring that at least one component, either the protein or the peptide, has low similarity to the training set. |

The table describes the dataset names, sizes, types, presence of natural peptides, and specific descriptions.

**Table 2 | Summary statistics (count, mean, standard deviation, minimum, 25th percentile, median, 75th percentile, and maximum) of protein-peptide sequence identity products for the five evaluation datasets after applying a 70% protein-peptide sequence identity threshold relative to the training set**

| Source | Count | Mean | Std | Min | Quantile-25 | Median | Quantile-75 | Max |
|---|---|---|---|---|---|---|---|---|
| LEADS_PEP_70% | 33 | 0.22 | 0.21 | 0.00 | 0.02 | 0.16 | 0.36 | 0.67 |
| Latest_70% | 75 | 0.12 | 0.11 | 0.00 | 0.04 | 0.12 | 0.16 | 0.60 |
| ncAA-1_70% | 138 | 0.16 | 0.23 | 0.00 | 0.02 | 0.05 | 0.19 | 0.70 |
| PFPD_70% | 20 | 0.14 | 0.15 | 0.00 | 0.02 | 0.08 | 0.22 | 0.57 |
| SinglePPD_Test_70% | 118 | 0.24 | 0.21 | 0.00 | 0.05 | 0.20 | 0.35 | 0.68 |

multi-scale topological and geometrical features using the PCL. These features capture intrinsic topological characteristics (such as the number and persistence of cavities and loops in the interface) and structural shape evolution across spatial scales, enabling the ML model to accurately predict DockQ scores (p-DockQ) that reflect interface quality. Detailed information on feature generations and model training is provided in the "Methods" section.

To reduce redundancy and the risk of potential data leakage from closely related sequences and corresponding complex structures in the evaluation datasets, we evaluate peptide-protein complexes predicted by AF2-M across five datasets filtered at a 70% sequence identity threshold. The datasets are labeled with "(70%)" in their names, such as LEADSPEP_70%, to indicate the sequence identity threshold applied. To comprehensively evaluate the p-DockQ score against the AF2-M and AF3 built-in confidence score (af_confidence), both AF2-M and AF3 confidence scores were used for ranking, and the top-1 model is selected

for further analysis. Models with a confidence score greater than 0.82 are considered positive predictions as this threshold yields the highest F1 score (see Supplementary Fig. 4), indicating that the model is predicted to be high-quality. Meanwhile, models with an actual DockQ score above 0.8 are classified as true high-quality structures, following established evaluation criteria[24,40,41].

As anticipated, the FPR is effectively reduced across all datasets (with 70% sequence identity cutoff) (Fig. 2A), and precision is greatly enhanced (Fig. 2B). TopoDockQ achieves reductions in FPR of up to 0.33 (LEADSPEP_70%) and precision gains of up to 0.17 (Test_70%). Additionally, the true positive rate (recall) and F1 score remain at satisfactory levels (Supplementary Fig. 5), with F1 score changes ranging from a decrease of 0.08 for Latest_70% to an increase of 0.09 for SinglePPD-Test_70%. The observed reduction in recall is consistent with a precision-recall trade-off, and reflects the method's emphasis on effectively filtering out false positives while maintaining overall performance.

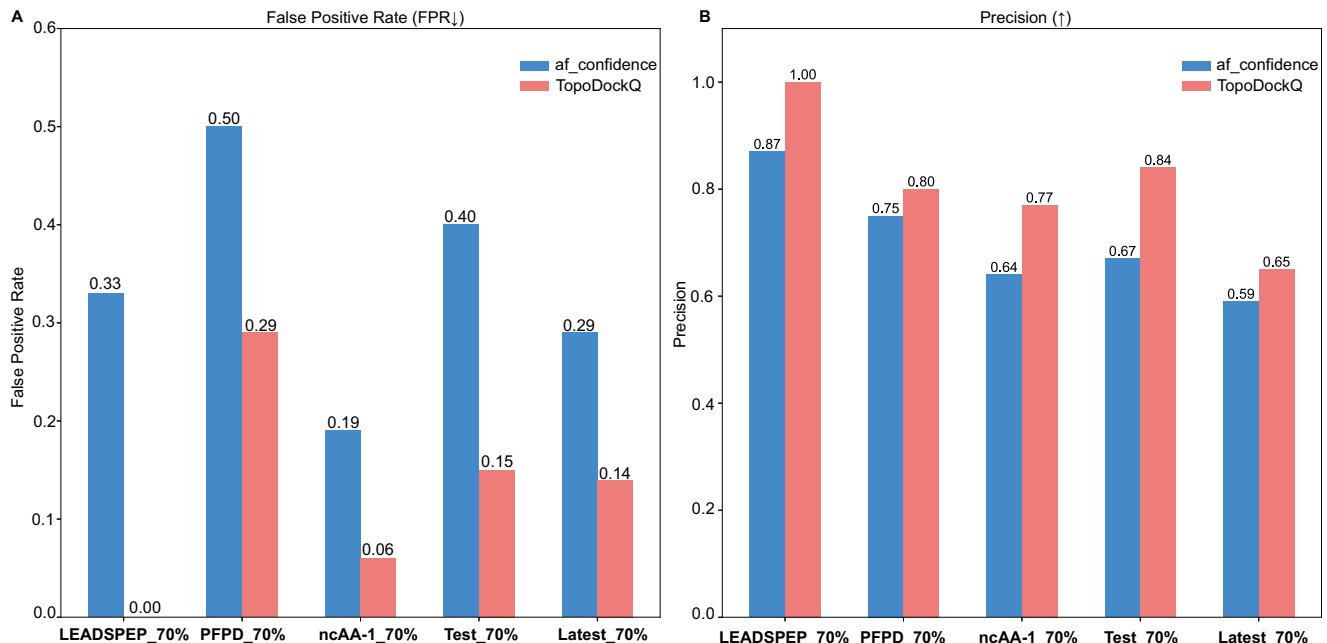

**Fig. 2 | Comparison of false positive rate and precision of Top-1 model selection with 70% protein-peptide sequence identity cutoff. A** The comparison of the False Positive Rate of the top-1 model selected by two confidence scores over five evaluation datasets with 70% protein-peptide sequence identity cutoff. **B** The

comparison of the Precision of the top-1 model selected by two confidence scores over five evaluation datasets with a 70% protein-peptide sequence identity cutoff. (up-arrow: higher value, better performance; down-arrow: lower value, better performance).

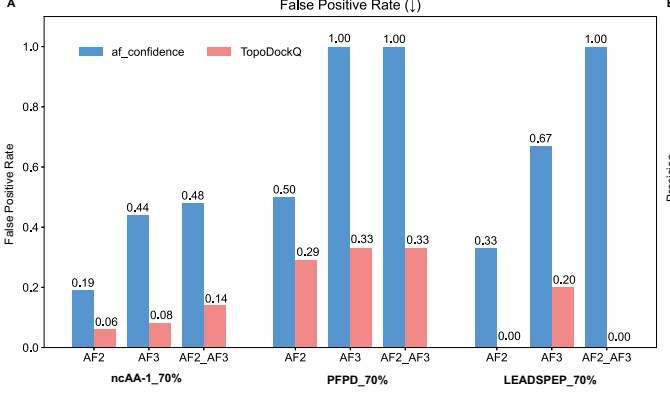
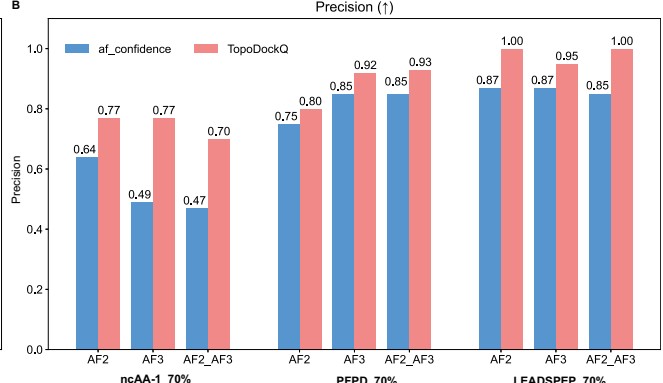

**Fig. 3 | Comparison of false positive rate and precision for Top-1 model selection using af_confidence and p-DockQ across different datasets for AF2-M and AF3-generated complexes. A** False Positive Rate (FPR) comparison between af_confidence (blue) and TopoDockQ (red) across ncAA-1, PFPD, and LEADSPEP datasets with a 70% protein-peptide sequence identity cutoff, demonstrating a consistent

reduction in false positives with TopoDockQ. **B** Precision comparison between af_confidence and TopoDockQ across the same datasets with a 70% protein-peptide sequence identity cutoff, highlighting an overall improvement in precision when using TopoDockQ for model selection.

To further evaluate model generalizability, we applied the same analysis to a filtered subset of SinglePPD_Test in which the product of protein and peptide sequence similarities to the training set is below 0.3. This dataset is named SinglePPD-Test_LowSimilarity. On this more challenging evaluation dataset, p-DockQ continues to substantially reduce the FPR and enhance precision and F1 score (see Supplementary Fig. 6, with only a slight decrease in TPR that remains at a relatively high level, indicating that the model's performance is not due to memorization of training data but rather reflects genuine generalization to novel peptide-protein pairs. Also, we utilized AF3 to generate complexes and conducted evaluation across ncAA-1_70%, PFPD_70%, and LEAD-SPEP_70% datasets. When evaluated AF3-generated complexes alone or in combination with AF2-M-generated complexes, the results consistently showed notably reduced FPR and improved precision (Fig. 3A, B).

To provide a full account of the evaluation of these datasets, we also report model performance for all metrics on the corresponding full datasets (i.e., without applying the sequence identity cutoff). These results are presented in Supplementary Figs. 2, 3, and 7, showing FPR, precision, TPR, and F1 metrics for both AF2- and AF3-generated complexes, respectively, and the observed performance is consistent with that reported for the filtered datasets, further supporting the robustness of the method.

## p-DockQ-driven scaffold selection for non-canonical peptide engineering

The p-DockQ score demonstrates greatly improved performance on the non-canonical peptide dataset (ncAA-1) where the ncAAs are substituted with natural amino acids during peptide-protein complex prediction. In this study, the canonical amino acid substitutions are based on those provided by the authors of the original crystal structures, where each ncAA is matched to

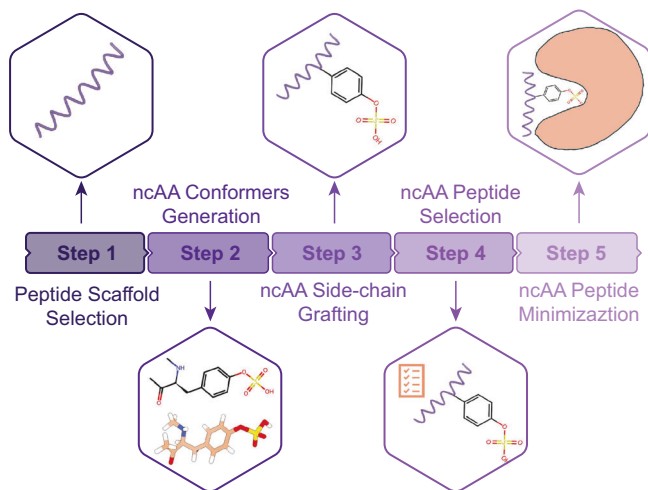

**Fig. 4 | The workflow of non-canonical peptide generation (ResidueX).** Workflow of ResidueX for ncAA incorporation into peptide scaffolds as five steps. (1) Peptide scaffold selection: A high-confidence peptide-protein structure is selected based on a high predicted DockQ (p-DockQ) score from TopoDockQ. (2) ncAA conformers generation: Up to 200 conformers of the desired ncAA, capped with methyl groups for stability, are generated from its SMILES representation. (3) ncAA side-chain grafting: The ncAA side-chain conformers are aligned and integrated into the peptide structure while preserving the backbone atoms (N, CA, C=O). (4) ncAA peptide selection: The p-DockQ scores of the generated ncAA peptides are predicted by TopoDockQ using the TopoDockQ model, and the highest-scoring conformer is selected. (5) ncAA peptide minimization: The selected ncAA peptide undergoes minimization with the protein part as the protocol based on Open Force Field and OpenMM.

its corresponding parent amino acid as annotated in the deposited PDB entry, if available. This strategy is chosen to maintain consistency with the empirical design and chemical reasoning reflected in the original annotations provided by the authors of the deposited crystal structures. On the ncAA-1 dataset, the p-DockQ scores predicted by TopoDockQ reduce the FPR for AF2-M-generated models from 0.20 to 0.06 (Supplementary Fig. 2A). Further analysis of AF3-predicted models reveal the most substantial improvement, with the FPR reduces from 0.44 to 0.07 (Supplementary Fig. 3A) and precision increases from 0.54 to 0.85 (Supplementary Fig. 3B) for peptide-protein complexes. Thus, the AF3-generated peptide-protein complexes are chosen for the study of scaffold selection for non-canonical peptide generation. After ranking and model selection, 41 models from the ncAA-1 dataset were selected, with 85% of these models having actual DockQ scores higher than 0.8. The high-quality of these selected docked models indicates their potential as scaffolds for generating non-canonical peptides, as the overall interface structures have been accurately predicted. These findings suggest that p-DockQ may be used to identify high-quality peptide-protein complex models from AF2-M and AF3 predictions when experimental structures of natural peptides are not available for use as scaffolds in the ncAA peptides design.

### p-DockQ-guided selection of ncAA peptide conformers in ResidueX workflow

ResidueX is a computational framework designed to enable structure-based design of non-canonical peptides by integrating peptide-protein complex models with user-defined ncAAs. While existing methods for incorporating ncAAs into peptide structures have enabled significant progress, they often rely on commercial software or require manual parameterization for force field compatibility. ResidueX provides a complementary and automated alternative that supports residue substitution directly from SMILES representations, without the need for user-defined force field parameters.

Structure-based ncAA peptide design typically begins with a known peptide scaffold. In our workflow, scaffold selection is guided by predicted

DockQ (p-DockQ) scores from TopoDockQ, allowing the identification of high-confidence peptide-protein complexes generated by AF2-M or AF3. This strategy enables a fully automated pipeline capable of generating structurally plausible ncAA-containing complexes from sequence inputs and ncAA SMILES. Although ResidueX can accept any peptide-protein complex structure as input, for the purpose of evaluating its de novo design capability, we focus on examples generated by AF3.

The ResidueX protocol consists of five modular stages: (1) scaffold selection, (2) ncAA conformer generation, (3) side-chain grafting with backbone preservation, (4) structure scoring using p-DockQ, and (5) energy minimization via OpenMM[42] and the Open Force Field[43]. Implementation details of each step are described in the "Methods" section.

The selected scaffold peptides are then processed through our designed workflow ResidueX (Fig. 4), to generate ncAA peptide conformers. Subsequently, p-DockQ scores are predicted for each generated conformer based on their persistent topological Laplacian-based features. The conformer with the highest p-DockQ score is selected as the ncAA peptide conformer to be minimized with corresponding protein parts with Open Force Field and OpenMM as described in the method. The p-DockQ of generated ncAA peptide-protein complexes is also used to filter out low-quality models, meaning that any ncAA peptide-protein complex with a p-DockQ below 0.82 will not be considered high-quality. The cumulative distribution plots in Supplementary Fig. 8, illustrate the percentage of ncAA peptide-protein complexes achieving various RMSD thresholds for both entire peptide structures (panel A) and ncAA side-chain conformations (panel B), and details for each threshold can be found in Supplementary Tables 1 and 2. These results indicate that p-DockQ, though developed with natural peptide-protein complexes, can be extended to reliably assess structural quality in complexes containing non-canonical peptides, thereby broadening its applicability in peptide design workflows. As shown in Supplementary Fig. 8, with the incorporation of p-DockQ to filter out low-quality models, 81.82% of the selected final ncAA peptide conformers have entire peptide RMSD values within 3 Å compared to their crystal structures.

### Considerations and challenges

Although the current TopoDockQ model performs well, some false positives still persist for both natural and non-canonical peptides, as neither af_confidence nor p-DockQ can fully eliminate incorrect predictions. In particular, variability in ncAA residue side-chain RMSD highlights several contributing factors. First, the absence of crystal water molecules in predicted models removes key stabilizing interactions, as water bridges can significantly influence peptide-protein binding. Second, organic molecules, co-factors, or metal ions present in crystal structures may alter protein folding and interaction patterns, leading to deviations in predicted models. Third, pH and protonation states impact both model selection accuracy and structural deviations, as side-chain charge states influence binding affinity, interaction stability, and final conformations. Our study assumes a physiological pH of 7.4, but variations in solution conditions can introduce unexpected electrostatic effects, altering both peptide-protein binding and ncAA side-chain positioning.

In addition, there is still room for improvement in selecting optimal conformers for peptides containing ncAAs, as these were not included in the training data. Incorporating ncAA-containing peptides into the training set may enable the model to learn distinct structural and interaction patterns specific to non-canonical side chains. Additionally, the lower proportion of ncAA residue side-chain RMSD values within 3 Å (approximately 50%) may be explained by the fact that p-DockQ is a comprehensive confidence score assessing the entire peptide-protein interface, whereas ncAA residue side-chain positioning represents only a subset of the interface. As part of our future work, we plan to augment the training set with peptides containing ncAAs. Our approach will begin with AlphaFold-predicted structures based on natural peptide sequences, followed by the in silico introduction of ncAA side chains to generate corresponding conformers with ResidueX. To address potential steric clashes or strain caused by non-canonical substitutions, we intend to apply a minimization step to refine

these structures and better capture their interfacial geometry. In parallel, we aim to adapt the DockQ scoring algorithm to accommodate complexes with ncAA residues, as current implementations are limited to canonical amino acids. The incorporation of ncAA-containing peptides into the training set is expected to enhance TopoDockQ's ability to recognize and select optimal conformers for peptides containing ncAAs.

Beyond ncAAs, another key structural class of interest is cyclic peptides, which present unique challenges for interface modeling. In addition to the absence of ncAAs in the training data, the training and evaluation datasets in this study consist entirely of linear peptides. TopoDockQ can also be applied to cyclic peptide-protein complexes if interface features are generated using the same protocol described in the "Methods" section. However, performance on cyclic peptides remains to be systematically evaluated and represents an important future direction.

Further improvements may also be achieved by addressing structural limitations that affect ncAA side-chain positioning. One contributing factor is that natural amino acids are initially used as placeholders during structure prediction, which can lead to backbone conformations that are not well suited to accommodate ncAAs. Since non-canonical side chains are often bulkier, energy minimization alone may be insufficient to resolve local strain, especially if protein backbone deviations from crystal structures further affect side-chain positioning. Additional refinement using molecular dynamics (MD) simulations could help relax both peptide and protein backbones along with ncAA side-chains to improve structural adaptation.

## Methods

### Dataset

This section summarizes all the datasets we used in this work. Table 1 gives an overview of datasets, including subsets for training (SinglePPD-Training), validation (SinglePPD-Validation), and evaluation (SinglePPD-Test, SinglePPD-Test_LowSimilarity, Latest, PFPD, PFPD_42, LEADSPEP, and ncAA-1 Datasets). The "Natural peptide" column indicates whether the peptides are naturally occurring, and detailed descriptions highlight the derivation sources and properties, including specific subsets with ncAAs. The peptide length distribution, summarized using boxplots, is shown in Supplementary Fig. 9. There is no overlap in PDB IDs between the training set and any of the validation or test datasets.

**SinglePPD dataset**. This dataset is derived from BioLip[38], a semi-manually curated database catering to biologically relevant protein-ligand interactions. In this study, the peptide-protein complexes are selected from the BioLiP database (2022 version). Only X-ray crystal structures with a resolution ≤ 2.5 Å are included. Our study exclusively considers natural peptides, maintaining a peptide-protein ratio of 1:1. Peptides containing ncAAs, cyclic peptides, and those covalently linked to protein components are excluded. Additionally, peptides interacting with at least two proteins in the crystal structure are removed. After these exclusions, 2221 unique Protein Data Bank (PDB) IDs remain, excluding duplicates present in datasets used for evaluation. The SinglePPD dataset was randomly split at the PDB ID level into training, validation, and test sets with sizes of 1837, 193, and 191, respectively.

**SinglePPD-Test_LowSimilarity dataset**. To further assess model generalization, we derived a subset of the SinglePPD-Test dataset by filtering complexes based on sequence similarity to the training set. Specifically, we retained only those test complexes for which the product of protein and peptide sequence similarities was below 0.3, ensuring that at least one component (protein or peptide) was sufficiently dissimilar to the training data. Protein sequence similarity was computed using global alignment with the BLOSUM62[44,45] substitution matrix, while peptide similarity was evaluated against the peptide paired with the most similar protein in the training set. This filtering resulted in 80 complexes, which we refer to as the SinglePPD-Test_LowSimilarity dataset (see Github Repo for details[46]).

**Latest dataset**. To evaluate the prediction performance of AF2-M without potential overlap with its training set, peptide-protein complexes from January 1, 2023, onward are collected and filtered according to the criteria established for SinglePPD. To ensure consistency with the peptide length distribution in SinglePPD (see Supplementary Fig. 9), 89 PDB entries were selected comprising single protein-single peptide complexes, each with peptides of 30 residues or fewer.

**PFPD dataset**. This benchmark dataset contains 27 non-redundant peptide-protein complexes used in the study of PIPER-FlexPepDock[19]. It has also been employed for peptide docking using various existing peptide docking methods.

**PFPD_42 dataset**. This benchmark dataset contains 42 redundant peptide-protein complexes used in the study of PIPER-FlexPepDock[19]. It is used for exploring the DockQ scores, L-RMSD, interface RMSD and Fnat distribution and correlation in this study.

**LEADSPEP dataset**. As the inaugural entry of the Lessons for Efficiency Assessment of Docking and Scoring (LEADS)[47] series, LEADS-PEP acts as a representative dataset. It is a publicly accessible benchmark dataset conceived to assess the proficiency of docking programs in mimicking peptide binding conformations, and to enable comparisons across diverse methods and parameters. LEADS-PEP carefully curated set consists of 53 peptide-protein complexes, with around five to six complexes chosen for each peptide length (from three to twelve residues) to ensure diversity concerning peptide features such as acidity, basicity, hydrophobicity, hydrophilicity, and aromaticity. In this study, to maintain consistency with single protein-single peptide docking, 43 out of the 53 complexes from the original benchmark were selected for evaluation. We refer to this subset as the LEADSPEP Dataset.

**ncAA-1 dataset**. During the filtering process of the SinglePPD dataset, we identified peptide-protein complexes that contain exactly one non-canonical amino acid (ncAA). After excluding complexes with cyclic peptides, covalent bonds between peptide and protein, interactions involving multiple proteins, and any structures with missing residues, we obtained 150 unique peptide-protein complexes. This curated subset forms the ncAA-1 dataset used in our study.

**Datasets with 70% sequence identity cutoff**. To evaluate model performance under reduced sequence similarity and mitigate potential data leakage from closely related training examples, we derived "70%" variants of five evaluation datasets: LEADSPEP, Latest, ncAA-1, PFPD, and SinglePPD-Test. In each case, peptide-protein complexes were retained only if the product of their protein and peptide sequence similarities to the training set was ≤ 70%, ensuring that at least one component (protein or peptide) in each complex is relatively dissimilar to the training data. A 70% sequence identity threshold is widely used to reduce redundancy in benchmarking datasets for both protein and peptide sequences[48–51], as it provides a practical balance between reducing redundancy and maintaining meaningful diversity, with sequence functionality shown to diverge more frequently below that level[52–54]. Protein sequence similarity was computed using global alignment with the BLOSUM62 substitution matrix, and peptide similarity was evaluated against the peptide paired with the most similar protein in the training set. All structural and curation criteria from the parent datasets were preserved.

- **LEADSPEP_70%**: Derived from the LEADSPEP dataset, resulting in 33 complexes that retain the curated diversity of the original benchmark while meeting the 70% similarity threshold.
- **Latest_70%**: Derived from the Latest dataset (peptide-protein complexes deposited after January 1, 2023), yielding 75 complexes with reduced sequence similarity while preserving recency and diversity.

**Table 3 | Cumulative percentage coverage of the five evaluation datasets and the SinglePPD-validation set at different protein-peptide sequence identity thresholds relative to the training set**

| Dataset | ≤10% | ≤20% | ≤30% | ≤40% | ≤50% | ≤60% | ≤70% | ≤80% | ≤90% | Total PDB IDs |
|---|---|---|---|---|---|---|---|---|---|---|
| LEADS_PEP | 34.88% | 44.19% | 51.16% | 62.79% | 69.77% | 69.77% | 76.74% | 81.40% | 86.05% | 43 |
| Latest | 32.58% | 73.03% | 78.65% | 82.02% | 83.15% | 84.27% | 84.27% | 84.27% | 85.39% | 89 |
| ncAA-1 | 66.67% | 69.33% | 72.67% | 76.00% | 78.00% | 80.00% | 92.00% | 93.33% | 94.00% | 150 |
| PFPD | 44.44% | 51.85% | 62.96% | 70.37% | 70.37% | 74.07% | 74.07% | 81.48% | 85.19% | 27 |
| SinglePPD_Test | 19.90% | 31.41% | 41.88% | 50.26% | 53.40% | 55.50% | 61.78% | 64.40% | 73.30% | 191 |
| SinglePPD_Validation | 22.28% | 33.68% | 45.60% | 49.74% | 54.92% | 56.48% | 59.07% | 61.66% | 68.39% | 193 |

For each dataset, the percentage values represent the cumulative proportion of protein-peptide pairs whose sequence identity product (protein identity × peptide identity) to the closest training set pair is less than or equal to the specified threshold (10%, 20%, …, 90%). This analysis quantifies the degree of sequence overlap between the evaluation/validation datasets and the training set, enabling assessment of potential data leakage. The *Total PDB IDs* column indicates the number of complexes in each dataset before any identity-based filtering.

- **ncAA-1_70%** : Derived from the ncAA-1 dataset (peptides containing exactly one non-canonical amino acid), yielding 138 complexes filtered by the 70% threshold.
- **PFPD_70%** : Derived from the PFPD benchmark dataset used in the PIPER-FlexPepDock study, yielding 20 complexes while retaining its curated benchmark quality.
- **SinglePPD-Test_70%** : Derived from the SinglePPD-Test set, yielding 118 complexes, ensuring at least one binding partner has low sequence similarity to the training set.

Summary statistics for these datasets are provided in Table 2, confirming that all sets satisfy the intended sequence identity cutoff while maintaining sufficient diversity for robust performance evaluation. Table 3 provides cumulative percentage coverage on different datasets at different protein-peptide sequence identity thresholds relative to the training set. This table indicates the degree of sequence overlap between the evaluation/validation datasets and the training set, which enables assessment of potential data leakage.

### Persistent combinatorial Laplacian-based DockQ prediction model (TopoDockQ)

This section gives an introduction to PCLs[34,35]. For readers interested in a concise definition regarding simplex, simplicial complex, chain complex, and combinatorial Laplacians that are used to form PCLs, please check out the Supporting Information. For formal definitions, we refer to the literature in refs. [34,35,37,55–57].

**Persistent combinatorial Laplacians.** Both topological and geometric information can be derived from analyzing the spectra of $q$th-order combinatorial Laplacian (see Supporting Information S3). However, such information is inherently limited because it only captures static connectivity within a single simplicial complex. In real-world applications where shapes dynamically change (e.g., evolving molecular structures), this static snapshot often fails to provide sufficient information for machine learning analysis. To better encode these dynamic shape variations, PCLs (also called persistent spectral graphs) are proposed by creating a sequence of simplicial complexes induced by varying a filtration parameter.

First, we consider a filtration of simplicial complex $K$ which is a nested sequence of subcomplexes $(K_t)_{t=0}^m$ of the final complex $K$:

$$\emptyset = K_0 \subseteq K_1 \subseteq K_2 \subseteq \cdots \subseteq K_m = K. \quad (1)$$

Second, for each subcomplex $K_t$, we denote its corresponding chain group to be $C_q(K_t)$, and the $q$-boundary operator will be denoted by $\partial_q^t : C_q(K_t) \rightarrow C_{q-1}(K_t)$. For simplicity, $C_q^t$ denotes the chain group $C_q(K_t)$. Third, we introduce persistence to the combinatorial Laplacian

operator. We define the subset of $C_q^{t+p}$ whose boundary is in $C_{q-1}^t$ as $\mathbb{C}_q^{t,p}$,

$$\mathbb{C}_q^{t,p} := \{\beta \in C_q^{t+p} \mid \partial_q^{t+p}(\beta) \in C_{q-1}^t\}. \quad (2)$$

On this subset, we define the $p$-persistent $q$-boundary operator denoted by $\partial_q^{t,p} : \mathbb{C}_q^{t,p} \rightarrow C_{q-1}^t$. Its corresponding adjoint operator is $(\partial_q^{t,p})^* : C_{q-1}^t \rightarrow \mathbb{C}_q^{t,p}$. The $q$-order $p$-persistent Laplacian operator $\Delta_q^{t,p} : C_q^t \rightarrow C_q^t$ associated with the filtration is

$$\Delta_q^{t,p} = \partial_{q+1}^{t,p}\left(\partial_{q+1}^{t,p}\right)^* + \partial_q^{t*}\partial_q^t. \quad (3)$$

The matrix representation of $\Delta_q^{t,p}$ (i.e., *persistent combinatorial Laplacian*) in the simplicial basis is

$$\mathcal{L}_q^{t,p} = \mathcal{B}_{q+1}^{t,p}(\mathcal{B}_{q+1}^{t,p})^T + (\mathcal{B}_q^t)^T \mathcal{B}_q^t, \quad (4)$$

where $\mathcal{B}_{q+1}^{t,p}$ is the matrix representation of $\partial_{q+1}^{t,p}$.

The spectrum of $\mathcal{L}_q^{t,p}$ is

$$\text{Spec}(\mathcal{L}_q^{t,p}) = \{\lambda_{1,q}^{t,p}, \lambda_{2,q}^{t,p}, \cdots, \lambda_{N_q^t, q}^{t,p}\}, \quad (5)$$

where $N_q^t = \dim C_q^t$ is the number of $q$-simplices in $K_t$, and the eigenvalues are listed in the ascending order. The multiplicity of zero in the spectrum of $\mathcal{L}_q^{t,p}$ is the $q$th order $p$-persistent Betti number $\beta_q^{t,p}$, which counts the number of (independent) $q$-dimensional holes in $K_t$ that still exists in $K_{t+p}$. The relation can also be observed in

$$\beta_q^{t,p} = \dim \ker \mathcal{L}_q^{t,p} = \#0 \text{ eigenvalues of } \mathcal{L}_q^{t,p}. \quad (6)$$

In 3D real-world applications, we focus on the 0, 1, 2th-order persistent Laplacians as they depict the relations among vertices, edges, triangles, and tetrahedra[34].

### Persistent combinatorial Laplacian-based feature generation for peptide-protein complex.

In this section, we apply the PCLs to the peptide-protein system to obtain a comprehensive topological and geometric representation of the peptide-protein interaction. Given that carbon (C), nitrogen (N), and oxygen (O) atoms are commonly found in both proteins and peptides, we restrict our analysis to these atoms in this study. Let $E = \{C, N, O\}$ and denote set $\mathcal{A}_1$ and $\mathcal{A}_2$ A2 to be a set of {C, N, O} atoms of the protein and a set of {C, N, O} atoms of peptide, respectively. Here, $\mathcal{A} = \mathcal{A}_1 \cup \mathcal{A}_2$ with $\mathcal{A}_1 \cap \mathcal{A}_2 = \emptyset$. Moreover, $\mathcal{A}_1 = \bigcup_{E_i \in E} \mathcal{A}_1(E_i)$ and $\mathcal{A}_2 = \bigcup_{E_j \in E} \mathcal{A}_2(E_j)$. In this work, we construct two types of simplicial complexes based on these atom sets:

1. nine nested families of Rips complexes based on nine element-specific combinations between protein and peptide: {CC, CN, CO, NC, NN, NO, OC, ON, OO},
2. nine nested families of Alpha complexes based on nine element-specific combinations between protein and peptide: {CC, CN, CO, NC, NN, NO, OC, ON, OO}.

For the Rips complex, we consider 9 types of element-specific combinations by forming a bipartite graph on a peptide-protein complex system. For instance, we set the pairwise distance within $\mathcal{A}_1(E_i)$ and $\mathcal{A}_2(E_j)$ to infinity, indicating that there are no internal connections within atoms of protein or atoms of peptide. Furthermore, we can assign a solid sphere to each atom and gradually increase its radii ($r$). If a sphere in a protein interacts with a sphere in a peptide, we set the pairwise distance between these two spheres to be their Euclidean distance. Otherwise, we still set this distance to infinity. This setup generates a nested family of Rips complexes $K_d^{\text{rips}}(E_iE_j)$ based on $\mathcal{A}_1(E_i)$ and $\mathcal{A}_1(E_i)$, where $d = 2r$. Here, $d$ is the distance-based filtration parameter. Similarly, we can get 8 more Rips complexes for other element-specific combinations.

For the Alpha complex, we also consider 9 types of element-specific combinations between protein and peptide, but we form a complete graph compared to the bipartite graph in Rips complex. For example, we collect all $\mathcal{A}_1(E_i)$ and $\mathcal{A}_2(E_j)$ in one set $\mathcal{A}(E_iE_j)$ and build up an Alpha complex $K_d^{\alpha}(E_iE_j)$ with $d$ representing the distance filtration parameter.

With these two types of simplicial complexes, we can obtain their $q$th-order 0-persistent Laplacians and their corresponding spectra. Meaningful topological and geometric information are preserved in these spectra. Here, we introduce three methods to extract features: (1) eigenvalue statistics, (2) Betti-0 counts in bins, and (3) barcode statistics.

**Eigenvalue statistics**. The key method of feature generation is to extract statistics of eigenvalues obtained from persistent Laplacians constructed using Rips complexes $K_d^{\text{rips}}(E_iE_j)$ at a specific distance filtration $d$. Here, $E_i$ and $E_j$ represent element types ({C, N, O}) for proteins and peptides, respectively. Let $\mathcal{L}_0^{t,0}(E_iE_j)$ be the 0th-order 0-persistent Laplacians corresponding to $K_d^{\text{rips}q}(E_iE_j)$. For simplicity, we use $\mathcal{L}_0^t$ in the following sections. According to the Method section, the set of eigenvalues of $\mathcal{L}_0^t$ is $\text{Spec}(\mathcal{L}_0^t)$ and the multiplicity of zero is $\beta_0^t$ (i.e., Betti-0 number). We also denote the set of non-zero eigenvalues of $\mathcal{L}_0^t$ to be $\text{Spec}^\star(\mathcal{L}_0^t)$.

We collect 7 types of statistical value on the spectrum as part of our features, including $\text{sum}(\text{Spec}^\star(\mathcal{L}_0^t))$, $\text{min}(\text{Spec}^\star(\mathcal{L}_0^t))$, $\text{max}(\text{Spec}^\star(\mathcal{L}_0^t))$, $\text{avg}(\text{Spec}^\star(\mathcal{L}_0^t))$, $\text{std}(\text{Spec}^\star(\mathcal{L}_0^t))$, $\text{var}(\text{Spec}^\star(\mathcal{L}_0^t))$ and $\beta_0^t$. Here, $\text{sum}(\cdot)$, $\text{min}(\cdot)$, $\text{max}(\cdot)$, $\text{avg}(\cdot)$, $\text{std}(\cdot)$, and $\text{var}(\cdot)$ represent the summation, minimum, maximum, average, standard deviation, and variance of a set of numbers, respectively. In this work, we have 9 types of element-specific Rips complex $K_d^{\text{rips}}(E_iE_j)$ by varying the element types $E_i$ and $E_j$. Moreover, we set the distance filtration parameter $d$ to range from 2 Å to 10 Å with a step size of 0.25 Å. This parameterization generates 33 distinct distance-based filtration values. Therefore, this eigenvalue statistics method generates $7 \times 9 \times 33 = 2079$ features. In general, this method can describe bond lengths and non-covalent interactions in biomolecules[36,58].

**Betti-0 counts in bins**. This feature generation method is also based on Rips complexes $K_d^{\text{rips}}(E_iE_j)$. In this work, we set $d = 7$ and denote its $q$th dimension barcode as $\mathbf{B^q} = \{[b_\alpha^q, d_\alpha^q]\}_{\alpha \in I^q}$, where $b_\alpha^q$ and $d_\alpha^q$ are the birth and death positions of a Betti-0 bar on the filtration axis, and $I^q$ is an index set with $\|I^q\|$ equals to the number of $q$-simplex that forms the Rips complex. We further partition the filtration axis into $N$ bins and subsequently count the number of death positions ($d_\alpha^q$) that fall within each bin. This gives us $N$ features. In this work, we focus on $\mathbf{B^0}$ and split its filtration axis to 21 bins, the first bin spans the range from 0 Å to 2 Å, while the remaining 20 bins are evenly spaced from 2 Å to 7 Å with a bin size of 0.25 Å. Given that we have 9 Rips complex, we can obtain $21 \times 9 = 189$ features.

**Barcode statistics**. We focus on $\mathbf{B^0}$ from Rips complex $K_d^{\text{rips}}(E_iE_j)$, and $\mathbf{B^1}$ and $\mathbf{B^2}$ from Alpha complex $K_d^{\alpha}(E_iE_j)$ for feature generation. For $\mathbf{B^0}$, we define a set of all death positions as $D^0 = \{d_\alpha^0\}_{\alpha \in I^0}$ and 6 statistical values can be generated, including $\text{avg}(D^0)$, $\text{std}(D^0)$, $\text{max}(D^0)$, $\text{min}(D^0)$, $\text{sum}(D^0)$,

and $\|D^0\|$. As for $\mathbf{B^1}$, we define sets of all birth, death, and persistence as $B^1 = \{b_\alpha^1\}_{\alpha \in I^1}$, $D^1 = \{d_\alpha^1\}_{\alpha \in I^1}$, and $P^1 = \{d_\alpha^1 - b_\alpha^1\}_{\alpha \in I^1}$, respectively. Then, we collect 18 statistical values from these sets, including $\text{avg}(B^0)$, $\text{std}(B^1)$, $\text{max}(B^1)$, $\text{min}(B^1)$, $\text{sum}(B^1)$, $\text{avg}(D^1)$, $\text{std}(D^1)$, $\text{max}(D^1)$, $\text{min}(D^1)$, $\text{sum}(D^1)$, $\|D^1\|$, $\text{avg}(P^0)$, $\text{std}(B^1)$, $\text{max}(P^1)$, $\text{min}(P^1)$, $\text{sum}(P^1)$, $B_k^1$, and $D_k^1$, where $k = \text{argmax}(P^1)$. Here $\text{sum}(\cdot)$, $\text{min}(\cdot)$, $\text{max}(\cdot)$, $\text{avg}(\cdot)$, $\text{std}(\cdot)$, $\text{var}(\cdot)$, and $\text{argmax}(\cdot)$ represent the summation, minimum, maximum, average, standard deviation, variance, and index of the maximum value in a set of numbers, respectively. Similarly, we obtain 18 statistical values for $\mathbf{B^2}$. To summarize, we get $(6 + 18 + 18) \times 9 = 378$ barcode statistical features for all types of element combinations.

We concatenate all types of features mentioned above together. In total, we have 2646 PCL-based features that encode rich persistent topological changes and shape evolution information of a given peptide-protein complex.

**TopoDockQ: a topological deep learning (TDL) model to predict DockQ**. Figure 1 illustrates the workflow of our TDL model—TopoDockQ to predict DockQ values. The training, validation, and evaluation datasets contain various peptide-protein complex structures, which are described in the Dataset section. For each peptide-protein complex in the training set, we analyze the interface defined by a 10 Å cutoff distance between the protein and the peptide. This peptide-protein interface comprises all 3D coordinates of carbon, nitrogen, and oxygen atoms, which is essentially a point-cloud dataset. Based on the training dataset, we construct various topological complexes, such as alpha complexes and Rips complexes, and generate 9 combinations of element-specific subcomplexes. We then extract features using PCL-based feature generation, yielding 2079 eigenvalue statistics features, 189 Betti-0 counts in bin features, and 378 barcode statistics features. These features are concatenated into a 1D feature array, which serves as the input for training a multilayer perceptron (MLP) model. For validation and evaluation, we replicate the feature generation process and apply the trained MLP model to predict DockQ values. One example of how intra-chain and inter-chain points are treated in distance matrix calculation during persistent Laplacians is described in Supplementary Fig. 10.

Our MLP model consists of 4 hidden layers, each with 2048 neurons. The input layer dimension is set to 2646, representing the PCL features as described previously. Feature generation for a single protein-peptide interface requires, on average, about 3 seconds per complex using 8 CPU cores and 12 GB of RAM. Features are scaled based on the training set, and batch normalization is applied to all hidden layers to enhance convergence and stability. Xavier initialization is used to initialize the model weights, ensuring balanced gradients during training. Hyperparameters were screened based on validation RMSE on the SinglePPD_Validation set, and the configuration with batch size 512 and learning rate 0.0005 was selected. The model checkpoint corresponding to the lowest validation RMSE was saved and used for evaluation (see Supplementary Fig. 11). The final model was trained on an NVIDIA A100 GPU with a batch size of 512 and a learning rate of 0.0005. Training required approximately two hours, and the model corresponding to the epoch with the lowest validation RMSE on the SinglePPD_Validation set was saved and used for subsequent evaluation.

**Peptide-protein complex prediction via AlphaFold2-Multimer and AlphaFold3**
**Protein and peptide sequences**. FASTA files for the peptide-protein complexes are prepared from the PDB database using the deposited sequences rather than those in the crystal structures (PDB IDs can be found in Github Repo)[46]. The deposited sequences are chosen over the crystallographic sequences because residues are often missing in the actual crystal structures, making the deposited sequences more applicable to real-world scenarios. To enable structure prediction using AF2-M, which exclusively supports canonical amino acids, each non-canonical amino acid in the peptide sequence must be substituted with a chemically

or structurally analogous canonical counterpart in the input FASTA file. In this study, when processing the ncAA-1 dataset, we adopted the canonical amino acid substitutions as annotated by the authors of the original crystal structures. This approach ensures alignment with the empirical rationale and structural interpretations underlying the deposited PDB entries. For example, in PDB ID 1NT1, the non-canonical residue O-sulfo-L-tyrosine was represented by the canonical amino acid tyrosine.

**Peptide-protein complex prediction with AlphaFold2-multimer.** The prepared FASTA files are utilized for predictions in AF2-M. The model preset is configured as "multimer" to generate complexes, and MSA is produced with a full database. The argument "num_multimer_pre-dictions_per_model" is set to 10 to generate 50 models, while other parameters are kept at their default settings. Due to the high computational demand, ParaFold[59] is used to generate feature files only using CPUs, and the built-in flag in AF2-M is employed to conduct predictions on pre-computed features. All 50 models were successfully generated for each peptide-protein complex without failure.

**Peptide-protein complex prediction with AlphaFold3.** The protein and peptide sequences are utilized as input for complex generation using the AF3 webserver, yielding five structural models for each case. The resulting CIF files are subsequently converted to PDB format using Open Babel and subjected to energy minimization with Amber14 in an implicit water model, implemented via OpenMM.

**Built-in confidence score: AF confidence score.** The quality of structure predictions is primarily evaluated using the global super-position TM-score. AF2 calculates predicted template modeling scores (pTM) through pairwise error predictions. In AF2-M, pTM assesses the quality of the intrachain sections, while the interface pTM (ipTM) evaluates the quality of interchain sections. The AF confidence score (af_confidence) is derived as a linear combination of pTM and ipTM (Eq. (7)):

$$af\_confidence = 0.2 \times pTM + 0.8 \times ipTM. \quad (7)$$

Similarly, the AF3 confidence score (af3_confidence) incorporates additional penalties for steric clashes and a slight weighting for disorder (Eq. (8)):

$$af\_confidence = 0.8 \times ipTM + 0.2 \times pTM + 0.5 \\ \times fraction\_disorder - 100 \times has\_clash, \quad (8)$$

where fraction_disordered is a scalar in the range 0–1 that indicates what fraction of the prediction structure is disordered, as measured by accessible surface area; has_clash is a boolean value indicating whether a protein structure contains a significant number of atomic clashes, specifically when more than 50% of a chain's atoms are involved in clashes, or any chain has over 100 clashing atoms, as defined by AF3.

**Assessment of peptide docking**
**Root mean square deviation (RMSD).** In bioinformatics, the RMSD of atomic positions measures the average distance between the atoms of superimposed molecules. RMSD is denoted as

$$RMSD = \sqrt{\frac{1}{N} \sum_{i=1}^{N} (\delta_i)^2}, \quad (9)$$

where $N$ represents the number of atoms and $\delta_i$ indicates the Euclidean distance between the $i$th pair of corresponding atoms. In the calculation of peptide backbone RMSD, $N$ shows the number of backbone atoms in the peptide, while $\delta_i$ is the Euclidean distance between the $i$th pair of corresponding backbone atoms in the peptide. In the evaluation, the RMSD of the entire peptide and backbone peptide is assessed using distinct approaches.

The RMSD of the entire peptide includes all heavy atoms in the peptide structure, whereas the backbone atoms are considered in the backbone peptide RMSD. A threshold of 3 Å is commonly used as a success criterion for both backbone and all-heavy-atom RMSD in peptide-protein docking, ensuring consistency with and comparability to existing peptide docking methods. For the ncAA residue side-chain RMSD calculation, only the ncAA residues were selected, and the RMSD was computed using their side-chain heavy atoms, as these atoms were specifically generated during model construction.

**DockQ.** DockQ[40,41] serves as a comprehensive quality metric to evaluate and compare docking models. It is regularly employed in CAPRI[40] and other benchmarks. Before we move to the definition of DockQ, we introduce the scaled root mean square deviations (RMSD$_{scaled}$) first:

$$RMSD_{scaled}(RMSD, d_i) = \frac{1}{1 + (\frac{RMSD}{d_i})^2}. \quad (10)$$

Here, RMSD can be either LRMSD or iRMSD, which represent the ligand RMSD (also known as peptide backbone RMSD) and interface RMSD, respectively. In addition, $d_i$ is a scaling factor, and we use $d_1$ for LRMSD and $d_2$ for iRMSD. With the definition of RMSD$_{scaled}$, the DockQ is then defined as:

$$DockQ(F_{nat}, LRMSD, iRMSD, d_1, d_2) \\ = \frac{F_{nat} + RMSD_{scaled}(LRMSD, d_1) + RMSD_{scaled}(iRMSD, d_2)}{3}, \quad (11)$$

where $F_{nat}$ represents the fraction of native interfacial contacts, $d_1$ and $d_2$ are set to 8.5 Å and 1.5 Å respectively as in the reference paper. DockQ ranges from 0 to 1, where 0 means the model is totally wrong and 1 means the model perfectly matches the reference structure.

In this work, we adopt the original DockQ classification scheme[40,41], where the DockQ scoring thresholds are set as follows: 0–0.23 for docking failure, 0.23–0.49 for acceptable quality, 0.49–0.8 for medium quality, and 0.8–1 for high-quality. One case is chosen as an example for visualization (Fig. 5A), after performing peptide docking on AF2-M, the peptide-protein complex is superimposed to the reference protein structure first. The protein part backbone RMSD is 0.4 Å, and the peptide backbone RMSD is 3.8 Å. Residues in contact with the protein are shown in green dots, and non-contact residues in red dots The interface RMSD is 1 Å and the F$_{nat}$ is 1.0, indicating all native contacts residue pairs are recalled. The overall DockQ score is higher than 0.8, falling in the range of the high-quality class.

The correlation between DockQ and L-RMSD (Fig. 5B) shows that while L-RMSD increases, DockQ does not always decrease proportionally, indicating that DockQ captures additional structural information. By incorporating interface RMSD and native contact fraction, DockQ enables a more accurate evaluation of model quality, making it a more reliable metric for peptide-protein interaction analysis. Furthermore, the distributions of interface RMSD and fractions of native contacts (Fig. 5C, D) demonstrate that higher DockQ scores correspond to lower interface RMSD and higher fractions of native contacts, reinforcing its effectiveness in distinguishing high-quality peptide-protein interactions. The threshold of high-quality ensures that the interface RMSD remains within 1 Å and 90% of the native contacts are recalled (Fig. 5C, D). Such high accuracy guarantees meaningful peptide-protein interaction patterns, suitable for further analysis and a better understanding of the interaction mechanism. Moreover, a high-quality peptide-protein interface ensures that the peptide can serve as a reliable scaffold for non-canonical peptide design. For the calculation of iRMSD, we use the residues at the interface whose backbone is within 10 Å in the reference structures. As for the evaluation of F$_{nat}$, we use the residue pairs at the interface whose heavy atoms are within 5 Å.

**Structural comparison and assessment with DockQ.** Prior to the DockQ calculations, each complex is superimposed onto the reference

**Fig. 5 | Viability assessment of DockQ as a quality measure for peptide docking results: 50 models generated by AlphaFold2-Multimer on PFPD_42 dataset. A** DockQ vs peptide backbone RMSD used as docking results assessment. The predicted peptide-protein complex is superimposed on the reference protein structure first. Since the protein backbone RMSD is 0.4 Å, only the reference protein structure is shown here. The protein part is colored in gold and extracted from the crystal (PDB ID: 1OU8). The crystal (PDB ID: 1OU8) peptide and AlphaFold2-Multimer-predicted peptide are colored pink and cyan, respectively. The peptide backbone RMSD is 3.8 Å while the DockQ value is 0.835. **B** Scatter plot of DockQ vs peptide backbone RMSD (LRMSD), with color for each quality class: red (high-quality), green (medium-quality), yellow (acceptable-quality), blue (incorrect). The vertical line means the boundary of 3 Å. **C** The distribution of interface RMSD (iRMSD) by DockQ range is shown in box plots. DockQ was utilized to assess the interface quality of docked models. **D** The distribution of $F_{nat}$ by DockQ range is shown in box plots.

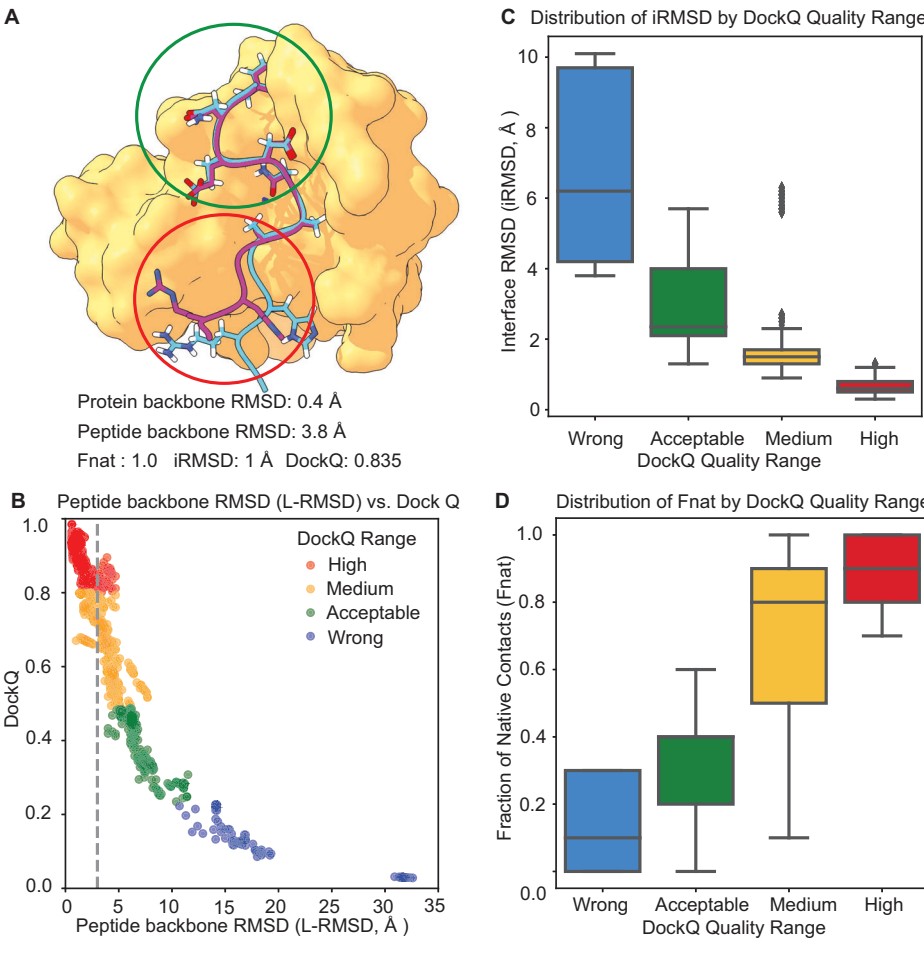

protein part first using the ProDy[60–62] package in Python. Only the residues and heavy atoms that are shared with reference complexes are preserved in each docked model, and the residues are renumbered to align with those in the reference. The calculation of DockQ, including LRMSD, iRMSD, and $F_{nat}$ are computed with pdb2sql[63,64].

## Non-canonical peptide generation

We have developed ResidueX (Fig. 4)[65], an RDKit-based package that enables the mutation of specific residues from natural amino acids to ncAAs without requiring manual parameterization. The process begins with the selection of a high-confidence peptide-protein structure, typically based on a high predicted DockQ (p-DockQ) score, followed by the definition of the desired ncAA using its SMILES string. ResidueX then generates up to 200 conformers of the ncAA, each capped with methyl groups for stability. The backbone atoms (N, CA, C=O) of the original natural residue in the scaffold are preserved, while the generated ncAA side-chain conformers are aligned and integrated into the peptide structure. The ResidueX workflow is designed to be flexible and modular. It can be applied to any peptide scaffold provided by the user, not limited to AlphaFold-generated peptides. Steps such as scoring, selection, and minimization are optional and can be adapted based on the specific modeling context. ResidueX provides a framework for incorporating ncAAs into peptide scaffolds, allowing users to adapt it to their specific modeling or design needs. If multiple ncAAs need to be incorporated into the peptide, they can conduct the sequential introduction of multiple non-canonical amino acid side-chains within the same peptide scaffold. For the generated ncAA peptides, their p-DockQ scores are predicted using our TopoDockQ model in the context of their corresponding protein partners. Finally, all ncAA peptide conformers are ranked based on

their p-DockQ scores, and the highest-scoring conformer is selected for further analysis.

## Minimization based on open force field and openMM

In our workflow, derived from Open Force Field Initiative example to treat post-modification cases, once the non-canonical reside is selected by residue name, "ACE" and "NME" capping will be done on the chosen residue first to be a capped ncAA. Then, the Sage force field will use am1bcc[66] charge generation method to assign partial charges on the capped ncAA. Only the ncAA residue part's assigned partial charged will be encoded into the charge library by the molecule SMARTS. The saved parameters will be assigned based on the matched chemical substructures. The partial charges of natural amino acid residues can be automatically assigned by Amber ff14sb force fields, and the ncAA residue in the peptide will be assigned accordingly by the encoded charges in the library. The generated ncAA peptide, along with the corresponding protein structure, will be minimized via mixed force fields with openMM[42].

## Data availability

The PDB files of generated peptide-protein complexes across multiple dataset used in this study are available at https://zenodo.org/records/15469415. The corresponding author can be contacted for additional requests.

## Code availability

The source code of the TopoDockQ model is available at https://github.com/XDaiNYU/TopoDockQ, reference number 46. The code of TopoDockQ feature generation is available at https://github.com/wangru25/TopoDockQ-Feature. The YouTube video that guides interested readers in

generating TopoDockQ features is available at https://youtu.be/fozzr7Y_bQc. A complete YouTube video guidance regarding feature generation, training, and inference is available at https://youtu.be/IGOtribF4u4. The package for ResidueX is available at https://github.com/XDaiNYU/ResidueX, reference number 65. The corresponding author can be contacted for additional requests.

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

## Acknowledgements
This work is supported by the U.S. National Institutes of Health (R35-GM127040). We thank NYU-ITS and the Simons Center for providing computational resources. R.W. is grateful for the support from the Simons Foundation and the Simons Center for Computational Physical Chemistry (SCCPC) at New York University. X.D. acknowledges partial support from a graduate fellowship from the Simons Center for Computational Physical Chemistry (SCCPC) at NYU.

## Author contributions
All authors conceived this work, and contributed to the original draft, review, and editing. X.-H.D. and R.W. conducted experiments, contributed to the development of TopoDockQ, ResidueX, and analyzed data. Y.-K.Z. provided supervision and resources and acquired funding.

## Competing interests
The authors declare no competing interests.
