## [Transparent Peer Review file · Communications Chemistry]

Topological Deep Learning for Enhancing Peptide-Protein Complex Prediction

Corresponding Author: Professor Yingkai Zhang

Version 0:

Reviewer comments:

Reviewer #1

(Remarks to the Author)

The manuscript describes TopoDockQ, a topological deep learning model designed to model selection in predicted protein-peptide complexes by predicting DockQ scores, and the ResidueX workflow designed to transform canonical residues into non-canonical amino acids (ncAA).

We agree with the authors on the importance of developing better methods to identify correct protein-peptide complexes, and the need to support non-native amino acids which are often encountered in therapeutic peptides. However, it seems that the presented approach has some flaws shedding doubt on the validity of the results as the ability to extrapolate beyond the training set has not been demonstrated.

The main concern is the creation of the data sets which seems to be done by selecting complexes based on deposition date in the PDB or by PDBID. This does not guarantee by any means that highly similar if not identical complexes are present in both the training set and the other sets. To demonstrate the ability to extrapolate beyond the training dataset and demonstrate learning rather than memorization the sets should have limited overlap in sequence and structure similarity. Furthermore, no timing information was provided in the manuscript for the calculation of features and the predicted DockQ Scores based on these features. It is also noted that the ResidueX workflow seems to be limited to a single modified amino acid side-chain. It is unclear whether that is a limitation of the method or not, but in real life cases there are often more than one modified amino acids in therapeutic peptides. The concept of grafting the modified side-chain post-docking assumes the peptide can dock in the proper conformation without the modified side-chain. It is unclear how often this is the case. Finally, while I was able to run feature generation example, I could not run the inference. These are the main reasons for which this manuscript is would need major modifications and clarifications before it can be considered for publication.

Below is a list of issues with the current manuscript:

The authors state that “best performances achieve success rates of 11.9% and 37.3% for top-1 and top-10 docked”. If this is for global docking it should be specified since conventional docking at a known binding site achieves higher success rates.

It is also stated that “Given that binding sites in most peptide docking tasks are not well-characterized or experimentally validated, global docking is generally more applicable.”. This statement is arguable, as often experimental or computational data can be used to identify binding pockets before docking peptides.

The authors state that “the significant conformational changes [20, 21] in both protein and peptide during binding, attributed to their high backbone flexibility, and the difficulties in accurately scoring and ranking docked poses [11]”. Both references 11 and 21 are 10 years old. Reference 21 is about conformational change when small molecules bind to proteins which tend affect the protein conformation more than highly flexible peptides.

The statement “MDockPeP2 improves global peptide docking performance, it still requires the unbound apo structure” is surprising. Most docking methods perform better with holo structures. Why is the apo structure required?

It is unclear what “This method achieves a success rate of ~90% for all heavy atoms in ncAA peptides” means.

I was unable to find a reference or description of the PFPD dataset?

It is unclear why a metric of 3Å backbone RMSD for the peptide is used? RMSD is not a good metric for peptides because it is global metric. It is used in small molecule docking because in this case most ligand atoms interact with the

macromolecule. For peptides this is often not the case and substantial subsets of peptide atoms do not interact with the macromolecule.

The authors mention that “After ranking and model selection, 41 models from the ncAA PEP dataset were selected, with 85% of these models having actual DockQ score higher than 0.8.” Does that mean that high quality docked peptides were selected with knowledge of the correct solution? It is also unclear what the 41 models correspond to. Is it for each protein-peptide pair? across the entire dataset?

The word “Figure” is missing before (5) in p-DockQ-guided section.

The authors state that “The 3Å cutoff is even accurate for small molecule docking tasks, so these generated ncAA peptides can provide meaningful insights on therapeutic peptide design and ncAA peptide-protein interaction analysis”. The convention for correct docking of small drug-like molecules is an RMSD of 2Å or less. An RMSD of 3Å on a modified amino acid side-chains atoms opens the door for creating the wrong atomic pairwise interactions.

I disagree with the statement “all-heavy atom RMSD remains the most relevant metric for evaluating model accuracy, particularly in ncAA peptide generation, as it provides a more holistic assessment of structural fidelity beyond side-chain-specific variations”. The authors actually demonstrate this point in Figure 4. This is also the reason what people use Fnat and DockQ to evaluate the quality of docked poses in protein-peptide and protein-protein docking.

The authors did not provide information of the amount of structural diversity they obtained in the 50 complexes generated with AF2m and how often they obtained 50 models.

The reason for defining Fnat using a 5Å rather than the 4Å default value is unclear.

Reviewer #2

(Remarks to the Author)

This manuscript presents the TopoDockQ model, designed to predict the DockQ score of protein–peptide complex structures generated by AlphaFold. The model demonstrates improved performance over AlphaFold's intrinsic confidence scores in identifying near-native structures from a pool of predicted models. Additionally, the authors introduce the ResidueX protocol, which enables the incorporation of non-natural amino acids into peptides based on user-defined specifications. Overall, both methods offer valuable tools for peptide design and could significantly contribute to the field.

However, a major issue with the manuscript lies in the inconsistent and, at times, confusing references to figures. Below are specific suggestions for improvement:

1. For improved readability and logical flow, consider first introducing the TopoDockQ model in the Results section, followed by its evaluation. The ResidueX protocol and its corresponding evaluation should then follow.
2. A brief overview of all datasets used in the study should be provided early in the Results section to help readers better contextualize the experiments.
3. Figures S1, S2, and S4 are not cited in the main text. Please ensure that all figures are appropriately referenced and discussed.
4. On page 3, the sentence “as high as 0.42 (Figure 1B)” appears before Figure 1A is discussed. To maintain the natural flow of figure references, consider switching panels A and B in Figure 1, so the text can instead read “as high as 0.42 (Figure 1A).”
5. On page 3, the text refers to “Figure 2A, B, and C,” but Figure 2C does not exist. Please correct the reference.
6. On page 3, “in false positives (1A and B)” should be revised to “in false positives (Figure 1A and B)” for clarity.
7. On page 3, the sentence “the false positive rate is effectively reduced across all datasets (Figure 2A), and precision is greatly enhanced (Figure 2B)” appears to incorrectly label the figure panels. Based on the figure content, Figure 2A seems to show precision, while Figure 2B shows false positive rate. Please verify and correct these references.
8. The statement “true positive rate (recall) and F1 score remained at satisfactory levels” on page 3 should reference Figure S4.
9. On page 4, the phrase “false positive rate from 0.20 to 0.06 for AF2-M generated models (Figure 2A)” should reference Figure 2B, not 2A.
10. It is recommended that the dataset names in Table 1 be matched exactly with those used in the figure captions and main text for consistency and clarity.
11. On page 12, the reference to “Supplementary Information (S6)” should be corrected to “Figure S6.”

Reviewer #3

(Remarks to the Author)

This manuscript describes two separate method developments. First is pDockQ which given a peptide protein complex - from alphafold - will predict the DockQ score. This is significant as DockQ relies rmsds from experimentally determined structures. This could be useful tool to assess the protein/peptide complexes without known structures. The statistics in figures 1 and 2 look reasonable though it is difficult to know how these will translate to truly new complexes. The only major limitation I see is that inability to handle cyclic peptides.

Second is ResidueX which can replace one of the standard 20 amino acids with a nonstandard amino acid in a peptide/protein complex using standard forcefield methodology. I think this is a fairly straightforward approach that could be done with a number of tools commercial or free.

Fundamentally this paper suffers from lack of readability. I had to read the manuscript 3 times before I understood the value of what is described in the paper. Primarily, a clear description of how this is all used together is used. Even now I am a bit hazy as to the exact workflow with noncanonical amino acids. The user must select a similar peptide with canonical amino acids and build the initial models using alphafold followed by ResidueX to change to the ncaas? If so how is the similar peptide selected.

The description of the persistent combinatorial Laplacian is distracting. I think it is very difficult to give a self-contained and concise description - for example the term abelian group is never defined. If there is something novel perhaps publishing as a separate paper in a journal dedicated to machine learning algorithms would make more sense.

Version 1:

Reviewer comments:

Reviewer #1

(Remarks to the Author)

The manuscript describes TopoDockQ, a topological deep learning model designed to predict the correctness of protein-peptide complexes interfaces by predicting DockQ scores. Additionally, the ResidueX workflow is described allowing to graft non canonical amino acids nCAAs onto peptides.

While the author made some efforts to address the concerns raised by the reviewers the manuscript remains extremely hard to read and seemingly disorganized making it difficult to assess the merits of the work. Several flaws are still present in the manuscript which would have to be re-worked profoundly, organized, and simplified before it could be considered for publication.

It is unclear whether the predicted DockQ scores would work for poses generated by conventional docking methods as it seems the training was limited to poses generated by Alpha Fold (AF). Sampling solutions is a well-known weakness AF as it often converges on a single solution. Why did the authors choose to use AF2 to generate complexes for training rather than including complexes generated by traditional peptide docking methods or modified version of AF2 modified specifically to enhance sampling ?

The result would be stronger if p-DockQ could improve the ranking of poses generated by conventional docking methods as well. The summary of the introduction claims "advancement of peptide-protein complex prediction by enhancing model selection through TopoDockQ" yet this is not demonstrated for conventional docking methods.

The ResidueX workflow is rather trivial and feels like a minor development, especially since the modified sidechains are not used during the pose prediction process.

Below is a laundry list of things to address in the manuscript:

First of all, there does not seem to be a natural flow in the manuscript. It would be more natural to have the introduction followed by the methods and datasets description so that when the results are discussed the reader is aware of the datasets and method referred to in the discussion.

Introduction

"GalaxyPepDock [12], performs similarity-based docking based on experimentally determined structures in the database" add PDB with a reference for "the database"

If AI methods for peptide docking methods are excluded from this paragraph then say so: "Conventional Peptide docking methods can be categorized ..."

The sentence about "PIPER-FlexPepDock [18] demonstrates: superior performance compared to other global docking algorithms" is only true if limited to conventional approaches as Deep Learning (DL) methods outperforms these methods.

Figure 1 is referenced on page 3 in the Introduction but the Figure only appears on page 5.

Methods

It is unclear how the singlePPD dataset was split into training validation and evaluation sets and how data leakage was prevented. The authors mentioned checking the complexes for similarity in the rebuttal but that is not mentioned here and feels to be an afterthought in response to the first reviewer's criticism. These datasets should be constructed properly from the beginning rather than checking after training whether there was data leakage.

The ncAA dataset comprises peptides with only 1 non-canonical amino acid. This is not representative the therapeutic peptides. Either fix the dataset or adjust the claims made in the manuscript.

The description of combinatorial Laplacian could largely be moved to the SI and replaced by a short summary of key points in the manuscript.

It is unclear what sequences were used for AF predictions. PDB files often provide sequences but 3D coordinates are not necessarily available for all the amino acids in the sequence, both for the protein and the peptide.

It is unclear on what complexes TopoDockQ was trained and what the overlap of the training set with the validation and test sets is.

Results and Discussion

Since the authors establish that DockQ is a better metric than RMSD for good docking solutions, why use the ligand RMSD in the results and discussion section?

The evaluation is made on a set of 27 complexes. This is a small dataset and there apparently no check for overlap with the training set.

NeuralPlexer is mentioned without any context in the middle of a paragraph about AF methods.

In the current manuscript the PPD, PFPD, LEADPEP, Latest and ncAA datasets are mentioned for the first time on page 3 in results and Discussion section without any reference, explanation of the acronym or summary of their content.

Calculating both protein and ligand RMSD on AF predictions is not a RESULT but a CONSEQUENCE of AF co-folding both sequences.

In the paragraph Improved Precision and Reduced False Positive Rate with p-DockQ Score: "In previous docking tests using the PFPD dataset" what does "previous docking" refer to?

"The p-DockQ score demonstrates greatly improved performance on the non-canonical peptide dataset (ncAA PEP)". I assume this means that selecting complexes using the p-DockQ score works better than using af_confidence? If my understanding is correct the ncAA containing peptides were predicted using natural amino acids. In this case why is it that p-DockQ performs so much better for this set than the 2 other sets? Is there a larger overlap with the training set for this dataset?

"accuracy remains limited because ncAA structures were not included in the training set" It is unclear why this would be the case. The approach assumes that the peptide is positioned properly while predicting the complex using canonical amino acids only. It is unclear to me that the few atoms added by the ncAA sidechain mutation are sufficient to affect the DockQ value?

I could not see an improvement in the software installation instructions.

In summary, there might be something interesting here about the p-DockQ score, however, the approach has several weaknesses that would have to be addressed before the results are believable and the manuscript needs a fundamental overhaul before it can be published.

Reviewer #2

(Remarks to the Author)

All my concerns have been addressed, and the issues raised by the other two reviewers have been satisfactorily resolved. I believe the manuscript is now ready for publication.

Version 2:

Reviewer comments:

Reviewer #1

(Remarks to the Author)

I thank the authors for their careful consideration of the previous critics and the modifications made to the manuscript which, as far as I am concerned, is now suitable for publication.

However, I still was unable to run the code and therefore have to take all results at face value, which is pretty disheartening given that this is now the third time this manuscript is getting to reviewers. How had can it be to go through the installation instructions and make sure they can be carried out by a potential user.

The Code Availability section provides the following link "The code of TopoDockQ feature generation is available at

<https://github.com/wangru25/TopoDockQFeature>." which does not exist.

I was able to get the conda environment setup but then the 02_tutorial_inference jupyter notebook is opened (assuming people know how to do that), it prominently displays a NOTE telling users to download a file called Ir_0.0005_bs_512.zip with no indication where to find it. After searching around for 20 minutes I gave up. This is rather upsetting given the repeated complaints from reviewers not being able to run the software to verify any of the results presented in the paper.

Authors' Response to Reviews of:
Topological Deep Learning for Enhancing Peptide-Protein
Complex Prediction (Manuscript ID:
COMMSCHEM-25-0234-T)

Xuhang Dai ^{1,*}, Rui Wang* ^{2,*}, and Yingkai Zhang ^{1,2,3,†}

¹Department of Chemistry, New York University, New York, NY 10003, USA

²Simons Center for Computational Physical Chemistry, New York University, New York, NY 10003, USA

³NYU-ECNU Center for Computational Chemistry, New York University Shanghai, Shanghai, 20062, CN

Dear Editor and Reviewers:

Thank you very much for your mail of 25-April-2025, and referees' reports concerning our manuscript entitled: "Topological Deep Learning for Enhancing Peptide-Protein Complex Prediction" (Manuscript ID: **COMMSCHEM-25-0234-T**). These comments are all valuable and helpful for revising and improving our paper, as well as providing important guidance to our researchers. We have accordingly gone through the referees' comments and have revised the manuscript. In the revised manuscript, the deleted text is marked in red with a strikethrough, while added text is shown in blue with a wavy underline. **We have also updated the order of several figures based on the reviewers' suggestions, and adjusted the figure numbers and their corresponding references throughout the manuscript accordingly.** Full responses to reviewers' comments are below (reviewers' comments are in bold black, while our responses are in blue).

Reviewer 1:

The manuscript describes TopoDockQ, a topological deep learning model designed to model selection in predicted protein-peptide complexes by predicting DockQ scores, and the ResidueX workflow designed to transform canonical residues into non-canonical amino acids (ncAA).

We agree with the authors on the importance of developing better methods to identify correct protein-peptide complexes, and the need to support non-native amino acids which are often encountered in therapeutic peptides. However, it seems that the presented approach has some flaws shedding doubt on the validity of the results as the ability to extrapolate beyond the training set has not been demonstrated.

(1). The main concern is the creation of the data sets which seems to be done by selecting complexes based on deposition date in the PDB or by PDBID. This does not guarantee by any means that highly similar if not identical complexes are present in both the training set and the other sets. To demonstrate the ability to extrapolate beyond the training dataset and demonstrate learning rather than memorization the sets should have limited overlap in sequence and structure similarity.

Answer-1: Thank you for your insightful comment. To assess model generalization, we filtered the SinglePPD.test set by comparing each test complex to the SinglePPD.train set and retaining only those for which the product of protein and peptide sequence similarities was below 0.3. This ensures that at least one component (protein or peptide) is dissimilar to training data, minimizing redundancy and potential data leakage. Protein sequence similarity was calculated using global alignment with the BLOSUM62 matrix, and peptide sequence similarity was assessed against the peptide paired with the most similar protein. The model performed consistently well on this subset, demonstrating the generalization of TopoDockQ. These results are now included in the Supporting Information Figure S7 and have been clarified in the revised manuscript in the Results and Discussion section under the subsection "Improved Precision and Reduced False Positive Rate with p-DockQ Score."

(2). Furthermore, no timing information was provided in the manuscript for the calculation of features and the predicted DockQ Scores based on these features.

Answer-2: We thank the reviewer for the helpful comment. We have now included timing information in the *Methods* section under the subsection "TopoDockQ: A Topological deep learning (TDL) model to predict DockQ". Training the model for 1000 epochs with a batch size of 512 on an NVIDIA A100 GPU took approximately 120 minutes. Feature generation for a single protein-peptide interface requires, on average, about 3 seconds per complex using 8 CPU cores and 12 GB of RAM.

(3). It is also noted that the ResiduesX workflow seems to be limited to a single modified amino acid side-chain. It is unclear whether that is a limitation of the method or not, but in real life cases there are often more than one modified amino acids in therapeutic peptides. The concept of grafting the modified side-chain post-docking assumes the peptide can dock in the proper conformation without the modified side-chain. It is unclear how often this is the case.

Answer-3: Thank you for your insightful comment. We would like to clarify that the ResiduesX workflow is designed to be applied iteratively, allowing for the sequential introduction of multiple non-canonical amino acid (ncAA) side-chains within the same peptide scaffold. To demonstrate this capability, we have included an example involving two ncAA side-chains in the Supporting Information. Regarding the assumption about the docking conformation, structure-based therapeutic peptide design commonly operates under the premise that the peptide backbone conformation remains largely stable upon side-chain modifications. This assumption is widely accepted in the field, as the primary goal is often to enhance binding affinity or specificity by optimizing side-chain interactions without altering the overall peptide conformation [1,2]. Our method aligns with this approach, providing a flexible framework for exploring side-chain modifications while maintaining the backbone structure. We have clarified these points in the revised manuscript.

(4). Finally, while I was able to run feature generation example, I could not run the inference. These are the main reasons for which this manuscript is would need major modifications and clarifications before it can be considered for publication.

Answer-4: Thank you for the comment. The trained model can be downloaded and dataset in the correct working directory for inference, as described in the Zenodo. We have clarified the inference setup in the revised Github and Zenodo websites, and we are available to assist if any further issues occur.

Below is a list of issues with the current manuscript:

1. **The authors state that “best performances achieve success rates of 11.9% and 37.3% for top-1 and top-10 docked”. If this is for global docking it should be specified since conventional docking at a known binding site achieves higher success rates.**

Answer: We thank the reviewer for the helpful comment. In the revised manuscript, we now explicitly specify that the reported success rates refer to global (blind) peptide–protein docking, where the binding site is not known in advance. We have revised the relevant sentence in the second paragraph of the Introduction section to improve clarity and provide appropriate context by distinguishing it from conventional site-specific docking. We also note that current global docking methods typically achieve success rates up to approximately 50%.

2. **It is also stated that “Given that binding sites in most peptide docking tasks are not well-characterized or experimentally validated, global docking is generally more applicable.”. This statement is arguable, as often experimental or computational data can be used to identify binding pockets before docking peptides.**

Answer: We thank the reviewer for the insightful comment. To address this, we have revised the statement to more accurately reflect the applicability of global docking. The updated version now reads: *“In cases where the binding site is not well-defined or experimentally validated, global docking provides a feasible approach for modeling peptide–protein interactions. It is also well-suited for investigating secondary or alternative binding sites beyond those identified a priori.”* in the second paragraph of the Introduction section.

3. **The authors state that “the significant conformational changes [20, 21] in both protein and peptide during binding, attributed to their high backbone flexibility, and the difficulties in accurately scoring and ranking docked poses [11]”. Both references 11 and 21 are 10 years old. Reference 21 is about conformational change when small molecules bind to proteins which tend affect the protein conformation more than highly flexible peptides.**

Answer: Thank you for the helpful feedback. We have revised the manuscript to clarify the discussion of the flexibility problem, which refers to the significant conformational changes in both peptide and protein molecules upon binding. We have also added more recent references that illustrate such conformational transitions. For example, in the reference paep [3] Figure 4b of the EphB4 receptor–ephrin-B2 antagonist complex (PDB ID: 2BBA), a beta-hairpin loop in the protein becomes disordered upon peptide binding. This and other updated cases underscore how peptide-protein interactions can involve considerable structural rearrangements on both sides.

4. **The statement “MDockPeP2 improves global peptide docking performance, it still requires the unbound apo structure” is surprising. Most docking methods perform better with holo structures. Why is the apo structure required?**

Answer: For pure global docking, MDockPeP2 needs an unbound apo structure for binding mode sampling first. The unbound structures are chosen in their study([4]; Binding Mode Sampling section). Any structure can be used in docking tasks. If the binding site is known, or we have prior

knowledge about the binding site, a holo structure will be more helpful. . To clarify this, we have revised the relevant sentence in the second paragraph of the Introduction section to read: "MDock-PeP2 still requires a protein structure as input, and in their study, the apo form was selected to enable site-independent global peptide-protein docking without prior knowledge of the binding site."

5. **It is unclear what "This method achieves a success rate of ~90% for all heavy atoms in ncAA peptides" means.**

Answer: We thank the reviewer for the helpful comment. To improve clarity and avoid ambiguity, we have removed the original sentence from the main text. A more detailed explanation is now provided in the Methods section under "Assessment of peptide docking", specifically in the "Root mean square deviation (RMSD)" subsection, where we clarify that the reported success rate is based on RMSD calculations over all heavy atoms in ncAA peptides.

6. **I was unable to find a reference or description of the PFPD dataset?**

Answer: We thank the reviewer for pointing this out. A reference (see [5]) and brief description of the PFPD dataset have been added to the Methods section under the "Dataset" subsection titled "PFPD: Benchmark dataset employed in PIPER-FlexPepDock".

7. **It is unclear why a metric of 3Å backbone RMSD for the peptide is used? RMSD is not a good metric for peptides because it is global metric. It is used in small molecule docking because in this case most ligand atoms interact with the macromolecule. For peptides this is often not the case and substantial subsets of peptide atoms do not interact with the macromolecule.**

Answer: We thank the reviewer for the helpful comment. Backbone RMSD is a widely accepted metric for evaluating peptide-protein docking accuracy, particularly in global docking scenarios where the peptide exhibits high flexibility and the binding site is not predefined. In our study, we adopt a 3.0Å backbone RMSD threshold as a practical and commonly used success criterion. This choice is consistent with previous studies. For example, MDockPeP applies a 3.0Å threshold for short peptides (less than 11 residues) and 4.0Å for longer ones [6], and PIPER-FlexPepDock reports performance at both 2Å and 3Å cutoffs [5]. To ensure consistency with established benchmarks and enable fair comparisons with existing methods, we have followed this standard practice in our evaluation. We have clarified this choice in the Methods section, under the "Root mean square deviation (RMSD)" subsection.

8. **The authors mention that "After ranking and model selection, 41 models from the ncAA PEP dataset were selected, with 85% of these models having actual DockQ score higher than 0.8." Does that mean that high quality docked peptides were selected with knowledge of the correct solution? It is also unclear what the 41 models correspond to. Is it for each protein-peptide pair? across the entire dataset?**

Answer: This process is p-DockQ driven process, so after sampling with the AlphaFold2 and AlphaFold3, for each peptide-protein complex case, if the p-DockQ is higher than a certain threshold, they will be selected into the downstream tasks with ncAA incorporation. The 41 models and their corresponding information can be found at: https://github.com/XDaiNYU/TopoDockQ/results/R_esidueX_ncAA_peptides.TopoDockQ_RMSD_results.csv. These are for the selected 41 models. The prediction or sampling capability of AlphaFold2-Multimer and AlphaFold3 is not the purpose in this study. We just use these as case studies for peptides with ncAAs. In real-world applications, known crystal structures can be used for this step. If there is no known structures, this a alternative choice to get a peptide scaffold. PFPD, Rosetta, and other methods could be chosen if AlphaFold cannot do the prediction with enough quality. We have clarified and reorganized this explanation in the revised Results and Discussion section under the subsection "p-DockQ-guided selection of ncAA peptide

conformers in the ResidueX workflow.”

9. **The word “Figure” is missing before (5) in p-DockQ-guided section.**

Answer: Thanks for pointing this out. We have fixed it in the revised manuscript.

10. **The authors state that “The 3Å cutoff is even accurate for small molecule docking tasks, so these generated ncAA peptides can provide meaningful insights on therapeutic peptide design and ncAA peptide-protein interaction analysis”. The convention for correct docking of small drug-like molecules is an RMSD of 2Å or less. An RMSD of 3Å on a modified amino acid side-chains atoms opens the door for creating the wrong atomic pairwise interactions.**

Answer: We agree that an RMSD of 2Å or less is conventionally used to define correct docking for small, drug-like molecules, where precise atomic interactions are critical. Our intention was not to suggest that a 3Å RMSD cutoff ensures perfect atomic-level accuracy for small molecule interactions, but rather to highlight that 3Å is a commonly accepted threshold in peptide docking tasks to assess overall binding mode similarity. In our context, 3Å RMSD serves to evaluate whether the overall peptide conformation is correctly predicted, including for ncAA-containing peptides. We have revised the manuscript to clarify this point and avoid any misunderstanding in the revised Results and Discussion section under the subsection “p-DockQ-guided selection of ncAA peptide conformers in the ResidueX workflow.”

11. **I disagree with the statement “all-heavy atom RMSD remains the most relevant metric for evaluating model accuracy, particularly in ncAA peptide generation, as it provides a more holistic assessment of structural fidelity beyond side-chain-specific variations”. The authors actually demonstrate this point in Figure 4. This is also the reason what people use Fnatt and DockQ to evaluate the quality of docked poses in protein-peptide and protein-protein docking.**

Answer: We thank the reviewer for the insightful comment. We agree that DockQ and Fnatt are widely used and informative metrics for evaluating protein-peptide and protein-protein docking. To avoid potential misunderstanding, we have removed the original sentence and clarified our rationale in the revised manuscript in the revised Results and Discussion section under the subsection “p-DockQ-guided selection of ncAA peptide conformers in the ResidueX workflow”. Specifically, due to the lack of standardized atom mappings for ncAA-containing peptides, DockQ and Fnatt are currently not applicable. In this context, we used all-heavy atom RMSD as a practical and consistent metric for assessing structural accuracy.

12. **The authors did not provide information on the amount of structural diversity they obtained in the 50 complexes generated with AF2m and how often they obtained 50 models.**

Answer: Thank you for your comment. For each peptide-protein complex, we successfully generated 50 structural models using AlphaFold2-Multimer, with no failures across all cases. To assess the structural diversity among the generated models, we examined the DockQ scores for each set of 50 models. Specifically, the lowest DockQ scores across all cases had a mean of 0.562 (min: 0.0038; max: 0.9811), while the highest DockQ scores had a mean of 0.771 (min: 0.0081; max: 0.9944). The details of the calculated DockQ values can be found in the `singlePPD_DockQ.csv` file within the `processed_data.zip` archive, available at: <https://zenodo.org/records/15469415>.

13. **The reason for defining Fnatt using a 5Å rather than the 4Å default value is unclear.**

Answer: 5Å is the standard criterion for Fnatt as defined in the DockQ score (see [7] here).

Reviewer 2:

This manuscript presents the TopoDockQ model, designed to predict the DockQ score of protein-peptide complex structures generated by AlphaFold. The model demonstrates improved performance over AlphaFold's intrinsic confidence scores in identifying near-native structures from a pool of predicted models. Additionally, the authors introduce the ResidueX protocol, which enables the incorporation of non-natural amino acids into peptides based on user-defined specifications. Overall, both methods offer valuable tools for peptide design and could significantly contribute to the field. However, a major issue with the manuscript lies in the inconsistent and, at times, confusing references to figures. Below are specific suggestions for improvement:

1. **For improved readability and logical flow, consider first introducing the TopoDockQ model in the Results section, followed by its evaluation. The ResidueX protocol and its corresponding evaluation should then follow.**

Answer: Thank you for the helpful suggestion. The structure of the Results and Discussion section has been revised to first introduce the TopoDockQ model and its evaluation, followed by the ResidueX protocol and its corresponding results. To improve clarity and coherence, we have added brief introductory paragraphs for both TopoDockQ and ResidueX, and adjusted the order of the associated figures accordingly.

2. **A brief overview of all datasets used in the study should be provided early in the Results section to help readers better contextualize the experiments.**

Answer: We appreciate this comment. A concise overview of all datasets used in this study has been added at the beginning of the Results and Discussion section. This addition is intended to clarify the specific purpose of each dataset and to help contextualize the subsequent analyses. A detailed description of each dataset remains in the Methods section for reference.

3. **Figures S1, S2, and S4 are not cited in the main text. Please ensure that all figures are appropriately referenced and discussed.**

Answer: Thanks for pointing this out. We have fixed these in the main text. Figure S1 has been cited in the 'Dataset subsection' for describing the peptide length statistical information. Figure S2 has been cited in the Method section-TopoDockQ hyperparameter selection and model training part. Figure S4 has been cited and discussed in the Results and Discussion section-'Improved Precision and Reduced False Positive Rate with p-DockQ Score' part, to show that F-1 score and precision score can still be kept at a relatively high level, or even improved when the FPR can be greatly reduced by TopoDockQ.

4. **On page 3, the sentence "as high as 0.42 (Figure 1B)" appears before Figure 1A is discussed. To maintain the natural flow of figure references, consider switching panels A and B in Figure 1, so the text can instead read "as high as 0.42 (Figure 1A)."**

Answer: We thank the reviewer for this helpful observation. To ensure a more logical and natural flow of figure references in the main text, we have switched the order of panels A and B in this Figure accordingly. The figure and corresponding text have been updated to reflect this change in the Results and Discussion section under the subsection "Improved Precision and Reduced False Positive Rate with p-DockQ Score."

5. **On page 3, the text refers to "Figure 2A, B, and C," but Figure 2C does not exist. Please correct the reference.**

Answer: We thank the reviewer for pointing this out. We have corrected the figure reference in the main text to "Figure 3A and B" (Figure 2A and B in the previous manuscript) to accurately reflect the

existing panels in the Results and Discussion section under the subsection “Improved Precision and Reduced False Positive Rate with p-DockQ Score.”

6. **On page 3, “in false positives (1A and B)” should be revised to “in false positives (Figure 1A and B)” for clarity.**

Answer: We appreciate the reviewer’s suggestion. We have revised the text to “Figure 2A and B” (Figure 1A and B in the previous manuscript) for clarity. This correction has been made in the Results and Discussion section under the subsection “Improved Precision and Reduced False Positive Rate with p-DockQ Score.”

7. **On page 3, the sentence “the false positive rate is effectively reduced across all datasets (Figure 2A), and precision is greatly enhanced (Figure 2B)” appears to incorrectly label the figure panels. Based on the figure content, Figure 2A seems to show precision, while Figure 2B shows the false positive rate. Please verify and correct these references.**

Answer: We thank the reviewer for carefully pointing out the figure reference inconsistencies. This has been fixed together with Questions 5 and 6.

8. **The statement “true positive rate (recall) and F1 score remained at satisfactory levels” on page 3 should reference Figure S4.**

Answer: We thank the reviewer for carefully pointing out the figure reference inconsistencies. This has been fixed together with Questions 5 and 6.

9. **On page 4, the phrase “false positive rate from 0.20 to 0.06 for AF2-M generated models (Figure 2A)” should reference Figure 2B, not 2A.**

Answer: We thank the reviewer for pointing this out. We have corrected the figure reference in the Results and Discussion section under the subsection titled “p-DockQ-driven scaffold selection for non-canonical peptide engineering”. Due to adjustments in figure order during revision, the referenced panel now appears as Figure 3B (Figure 2B in the previous manuscript), and the citation has been updated accordingly.

10. **It is recommended that the dataset names in Table 1 be matched exactly with those used in the figure captions and main text for consistency and clarity.**

Answer: We thank the reviewer for this helpful suggestion. To improve consistency and clarity, we have updated the dataset names in Table 1 to match those used in the main text and figure captions—specifically, standardizing to LEADSPEP, PFPD, ncAA, and Latest. The name SinglePPD_Test has been retained to distinguish it as a specific split of the SinglePPD dataset, and it is referenced as Test in the figures for brevity and readability. We believe this balance preserves clarity while maintaining consistency across the manuscript.

11. **On page 12, the reference to “Supplementary Information (S6)” should be corrected to “Figure S6.”**

Answer: We thank the reviewer for pointing this out. The reference has been corrected from “Supplementary Information (S6)” to “Figure S6” in the main text.

Reviewer 3:

This manuscript describes two separate method developments. First is pDockQ which given a peptide protein complex - from alphafold - will predict the DockQ score. This is significant as DockQ relies rmsds from experimentally determined structures. This could be useful tool to assess the protein/peptide complexes without known structures. The statistics in figures 1 and 2 look reasonable though it is difficult to know how these will translate to truly new complexes.

1. **The only major limitation I see is that inability to handle cyclic peptides.**

Answer: Thank you for pointing this out. Our model is specifically designed for linear peptide-protein complexes, and both training and evaluation were conducted exclusively on linear peptides. Cyclic peptides were not included in the dataset and have not been tested in this study. We have clarified this limitation in the revised manuscript, specifically in the "Considerations and Challenges" subsection under the Results and Discussion section.

2. **Second is ResidueX which can replace one of the standard 20 amino acids with a nonstandard amino acid in a peptide/protein complex using standard forcefield methodology. I think this is a fairly straightforward approach that could be done with a number of tools commercial or free.**

Answer: Thank you for your comment. ResiduesX was developed to simplify the process of substituting standard amino acids with non-canonical amino acids (ncAAs) in peptide-protein complexes. This workflow offers several advantages: it provides a user-friendly and straightforward approach that minimizes the need for extensive customization, ensures reproducibility across different projects, which is especially important for therapeutic peptide design. It is also compatible with standard force field methods, making it suitable for existing modeling workflows. We would like to emphasize that ResiduesX is intended to offer an additional option to facilitate structure-based design, rather than to compete with existing tools. We have clarified these points in the revised manuscript in the revised Results and Discussion section under the subsection "p-DockQ-guided selection of ncAA peptide conformers in the ResidueX workflow".

3. **Fundamentally this paper suffers from lack of readability. I had to read the manuscript 3 times before I understood the value of what is described in the paper. Primarily, a clear description of how this is all used together is used. Even now I am a bit hazy as to the exact workflow with noncanonical amino acids. The user must select a similar peptide with canonical amino acids and build the initial models using alphafold followed by ResidueX to change to the ncaas? If so how is the similar peptide selected.**

Answer: Thank you for the helpful suggestion. We have substantially revised and reorganized the *Results and Discussion* section to enhance clarity and logical flow. To enable structure prediction using AlphaFold2-Multimer, which exclusively supports canonical amino acids, each non-canonical amino acid in the peptide sequence must be substituted with a chemically or structurally analogous canonical counterpart in the input FASTA file. In this study, the canonical amino acid substitutions were based on those provided by the authors of the original crystal structures, where each ncAA was matched to its corresponding parent amino acid as annotated in the deposited PDB entry, if available. This strategy was chosen to maintain consistency with the empirical design and chemical reasoning reflected in the original annotations provided by the authors of the deposited crystal structures. For example, in PDB ID 1NT1, the non-canonical residue O-sulfo-L-tyrosine was represented by the canonical amino acid tyrosine. The section now begins with the introduction and evaluation of the TopoDockQ model, followed by a clear description of the ResidueX protocol and its associated results. To further improve readability, we have rewritten the introductory paragraphs for both components and adjusted the figure order to match the updated structure, providing a more coherent presentation of the overall workflow.

4. The description of the persistent combinatorial Laplacian is distracting. I think it is very difficult to give a self-contained and concise description. For example, the term abelian group is never defined. If there is something novel, perhaps publishing as a separate paper in a journal dedicated to machine learning algorithms would make more sense.

Answer: Thanks for the suggestion. We have carefully revised and shortened the math method section. These modifications provide more intuitive explanations with concrete examples and reduce math jargon while preserving essential mathematical rigor. We believe the revised methods section better aligns with the journal's interdisciplinary scope.

References

- [1] Justin M. Torner, Yuwei Yang, David Rooklin, Yingkai Zhang, and Paramjit S. Arora. Identification of secondary binding sites on protein surfaces for rational elaboration of synthetic protein mimics. *ACS Chemical Biology*, 16(7):1179–1183, 2021.
- [2] Ashley E. Modell, Frank III Marrone, Nihar R. Panigrahi, Yingkai Zhang, and Paramjit S. Arora. Peptide tethering: Pocket-directed fragment screening for peptidomimetic inhibitor discovery. *Journal of the American Chemical Society*, 144(3):1198–1204, 2022.
- [3] Tomer Tsaban, Julia K Varga, Orly Avraham, Ziv Ben-Aharon, Alisa Khramushin, and Ora Schueler-Furman. Harnessing protein folding neural networks for peptide–protein docking. *Nature communications*, 13(1):176, 2022.
- [4] Xianjin Xu and Xiaoqin Zou. Predicting protein–peptide complex structures by accounting for peptide flexibility and the physicochemical environment. *Journal of chemical information and modeling*, 62(1):27–39, 2021.
- [5] Nawsad Alam, Oriel Goldstein, Bing Xia, Kathryn A Porter, Dima Kozakov, and Ora Schueler-Furman. High-resolution global peptide-protein docking using fragments-based piper-flexpepdock. *PLoS computational biology*, 13(12):e1005905, 2017.
- [6] Xianjin Xu, Chengfei Yan, and Xiaoqin Zou. Mdockpep: An ab-initio protein-peptide docking server. *Journal of Computational Chemistry*, 39(28):2409–2413, 2018.
- [7] Sankar Basu and Björn Wallner. Dockq: a quality measure for protein-protein docking models. *PloS one*, 11(8):e0161879, 2016.

Authors' Response to Reviews of:
Topological Deep Learning for Enhancing Peptide-Protein
Complex Prediction (Manuscript ID:
COMMSCHEM-25-0234-T)

Xuhang Dai ^{1,*}, Rui Wang* ^{2,*}, and Yingkai Zhang ^{1,2,3,†}

¹Department of Chemistry, New York University, New York, NY 10003, USA

²Simons Center for Computational Physical Chemistry, New York University, New York, NY 10003, USA

³NYU-ECNU Center for Computational Chemistry, New York University Shanghai, Shanghai, 20062, CN

Dear Editor and Reviewers:

Thank you very much for your mail of 17-July-2025, and referees' reports concerning our manuscript entitled: "Topological Deep Learning for Enhancing Peptide-Protein Complex Prediction" (Manuscript ID: **COMMSCHEM-25-0234-T**). We have accordingly reviewed the referees' comments and revised the manuscript. In the revised manuscript, the deleted text is marked in red with a strikethrough, while added text is shown in blue with a wavy underline. **We have also updated the order of several figures based on the reviewers' suggestions. To further address concerns about potential data leakage and generalization, we evaluated TopoDockQ on five filtered datasets (LEADSPEP_70%, Latest_70%, ncAA-1_70%, PFPD_70%, and SinglePPD-Test_70%), each retaining only peptide-protein complexes with $\leq 70\%$ sequence identity to the training data, and have updated the reported results accordingly in the revised manuscript. Full responses to reviewers' comments are below (reviewers' comments are in bold black, while our responses are in blue).**

Reviewer 1:

The manuscript describes TopoDockQ, a topological deep learning model designed to predict the correctness of protein-peptide complexes interfaces by predicting DockQ scores. Additionally, the ResidueX workflow is described allowing to graft non canonical amino acids ncAAs onto peptides.

While the author made some efforts to address the concerns raised by the reviewers the manuscript remains extremely hard to read and seemingly disorganized making it difficult to assess the merits of the work. Several flaws are still present in the manuscript which would have to be re-worked profoundly, organized, and simplified before it could be considered for publication.

(1) It is unclear whether the predicted DockQ scores would work for poses generated by conventional docking methods as it seems the training was limited to poses generated by Alpha Fold (AF). Sampling solutions is a well-known weakness AF as it often converges on a single solution. Why did the authors choose to use AF2 to generate complexes for training rather than including complexes generated by traditional peptide docking methods or modified version of AF2 modified specifically to enhance sampling.

Answer-1: We thank the reviewer for raising this point. Our work is specifically motivated by a recognized challenge in AlphaFold2-based peptide-protein complex prediction, the issue of high-confidence but incorrect predictions (false positives) when relying solely on the built-in af_confidence scores. To address this, our research starts from improving model selection within AlphaFold-generated peptide-protein complexes by developing an external scoring system to distinguish the high-quality models during model selection process.

Our study focusing on AlphaFold-generated structures was also informed by benchmarking results. **Specifically, we compared the success rates of AlphaFold2 (AF2) and AlphaFold3 (AF3) on the same evaluation dataset against the reported results of several widely used global peptide docking methods. Both AF2 and AF3 consistently outperformed traditional methods in terms of generating high-quality peptide-protein complex structures.** As the reviewer also acknowledged in Question-4, deep learning-based methods such as AlphaFold2 substantially outperform traditional docking algorithms like PIPER-FlexPepDock, pepATTRACT, and HADDOCK on this task. Given that traditional methods struggle to generate conformations of high quality, we focus on AlphaFold2-generated models as the basis for our study for peptide-protein complexes generation. An additional advantage, as noted by the authors of PIPER-FlexPepDock [1], is that AF2 only requires sequences as inputs without the need for structural information, and it offers substantial practical benefits in speed, typically generating five models in around 20 minutes (using MMSeqs2 for MSA generation), compared to several hours required by traditional docking protocols such as PIPER-FlexPepDock.

In addition to AF2, we also evaluated our scoring model on complexes generated by AlphaFold3, which employs a different architecture and sampling strategy. The consistent performance of our model on AF3-generated structures suggests that it generalizes well to complexes generated by different docking approaches.

We sincerely appreciate this suggestion and agree that applying our model to conventional docking outputs would be a valuable direction for future exploration.

(2) The result would be stronger if p-DockQ could improve the ranking of poses generated by conventional docking methods as well. The summary of the introduction claims "advancement of peptide-protein complex prediction by enhancing model selection through TopoDockQ" yet this is not demonstrated for conventional docking methods.

Answer-2: We appreciate the reviewer's suggestion and agree that evaluating p-DockQ on conventional docking methods is an interesting future direction. The motivation of our research is the limitation of

the AlphaFold2 built-in confidence score (af.confidence) during peptide–protein complex generation and selection. In this study, we compare our method directly with af.confidence, aiming to reduce false positives and improve precision in selecting models of high-quality from AlphaFold-generated results. To clarify this focus, we have revised the summary sentence in the Introduction to: *“In summary, this study contributes to the advancement of peptide–protein complex prediction within the AlphaFold framework by enhancing the model selection step through TopoDockQ.”*

(3) The ResidueX workflow is rather trivial and feels like a minor development, especially since the modified sidechains are not used during the pose prediction process.

Answer-3: We thank the reviewer for this valuable comment and the opportunity to provide clarification. We would like to clarify that the modified sidechains (i.e., ncAA substitutions) are indeed used during the pose prediction process. The modified side chains are incorporated during ncAA peptide conformer generation, scoring, and selection. In the current canonical peptide–protein complexes are first predicted by AlphaFold. Those with high p-DockQ scores are selected as scaffolds, after which ncAA side chains are introduced to generate modified peptide conformers for further evaluation (See Section “p-DockQ-guided selection of ncAA peptide conformers in ResidueX workflow”).

ResidueX is intended to be *straightforward* and user-friendly. To the best of our knowledge, there is no existing package or software that allows ncAA incorporation without requiring customized force fields or additional parameterization, while also being fully compatible and automated with Python-based MD simulation packages (we use OpenMM here).

ResidueX is designed to be flexible and modular, capable of working with any peptide scaffold provided by the user, not only AlphaFold-generated peptides. Steps such as scoring, selection, and minimization are optional and can be tailored to the modeling context. The evaluation was conducted using ncAA-1(peptides with single ncAA) dataset cases, with AF3-generated models chosen for the case studies; the results confirmed the effectiveness of the workflow. ResidueX provides a framework for incorporating ncAAs into peptide scaffolds, allowing users to adapt it to their specific modeling or design needs.

Below is a laundry list of things to address in the manuscript:

1. **First of all, there does not seem to be a natural flow in the manuscript. It would be more natural to have the introduction followed by the methods and datasets description so that when the results are discussed the reader is aware of the datasets and method referred to in the discussion.**

Answer: Thank you for the suggestion. While the current structure follows the journal’s formatting guidelines, which require the Methods section to be placed after the Results, we have added a paragraph at the beginning of the Results and Discussion section to introduce the datasets and key related information from the Methods.

2. **Introduction: “GalaxyPepDock [12], performs similarity-based docking based on experimentally determined structures in the database” add PDB with a reference for “the database”.**

Answer: We thank the reviewer for this helpful suggestion. We have clarified in the revised manuscript that GalaxyPepDock performs similarity-based docking using experimentally determined structures from the PepBind database. A reference to the PepBind resource [2] has been added accordingly in the Introduction section.

3. **Introduction: If AI methods for peptide docking methods are excluded from this paragraph then say so: “Conventional Peptide docking methods can be categorized ...”.**

Answer: Thank you for the suggestion. We have revised the sentence as suggested: “Conventional peptide docking methods can be categorized ...” in this paragraph in the introduction section.

4. **Introduction: The sentence about “PIPER-FlexPepDock [18] demonstrates: superior performance compared to other global docking algorithms” is only true if limited to conventional approaches**

as Deep Learning (DL) methods outperforms these methods.

Answer: We thank the reviewer for the observation. To clarify that the comparison is made within traditional (non-deep learning) methods, we have revised the sentence in the Introduction section to begin with: “Among these traditional docking methods,”.

5. **Introduction: Figure 1 is referenced on page 3 in the Introduction but the Figure only appears on page 5.**

Answer: Thank you for pointing this out. We have adjusted the placement of Figure 1 to appear closer to its first reference in the Introduction.

6. **Methods: It is unclear how the singlePPD dataset was split into training validation and evaluation sets and how data leakage was prevented. The authors mentioned checking the complexes for similarity in the rebuttal but that is not mentioned here and feels to be an afterthought in response to the first reviewer’s criticism. These datasets should be constructed properly from the beginning rather than checking after training whether there was data leakage.**

Answer: We thank the reviewer for raising this important point. The SinglePPD dataset was split strictly at the **PDB ID level**, ensuring that no PDB ID assigned to the validation, test sets, or any other evaluations appears in the training set. To ensure comprehensive evaluation and avoid potential data leakage, we report in the main text results for evaluation datasets filtered at a protein–peptide sequence identity product $\leq 70\%$ against all sequences in the training set. In the revised manuscript, we present results for **five such datasets**: SinglePPD-Test_70%, PFPD_70%, LEADSPEP_70%, Latest_70%, and ncAA-1_70%. As summarized in Table 1 in the manuscript, the resulting subsets (mean values denote the average protein–peptide sequence identity product; ranges in parentheses) comprise LEADSPEP_70% (33 complexes; mean: 0.22, range: 0.00–0.67), Latest_70% (75; mean: 0.12, range: 0.00–0.60), ncAA-1_70% (138; mean: 0.16, range: 0.00–0.70), PFPD_70% (20; mean: 0.14, range: 0.00–0.57), and SinglePPD-Test_70% (118; mean: 0.24, range: 0.00–0.68). This filtering ensures that at least one binding partner (protein or peptide) in each complex is relatively dissimilar to the training data. A 70% sequence identity threshold is widely used to reduce redundancy in benchmarking datasets for both protein and peptide sequences [3–6], as it provides a practical balance between reducing redundancy and maintaining meaningful diversity, with sequence functionality shown to diverge more frequently below that level [7–9].

The results for these filtered datasets are presented in the **main text**, while the previous performance results on the full, unfiltered datasets have been moved to the **Supporting Information** for completeness. Importantly, the performance trends are consistent; the reduction in FPR and the improvement in precision remain substantial across the 70% datasets, confirming the robustness of the model under reduced sequence similarity. To further test generalization under more stringent conditions, we also constructed the **SinglePPD-Test.LowSimilarity(30%)** dataset, in which the product of protein and peptide sequence similarities is $\leq 30\%$. Even under this challenging setting, the model’s performance remains strong, demonstrating that TopoDockQ is not simply memorizing training data but generalizes well to novel protein–peptide pairs. These additions and clarifications have been incorporated into the **Results and Discussion** section of the revised manuscript, ensuring that the dataset construction process and our measures to prevent data leakage are clear and transparent to the reader.

7. **Methods: The ncAA dataset comprises peptides with only 1 non-canonical amino acid. This is not representative the therapeutic peptides. Either fix the dataset or adjust the claims made in the manuscript.**

Answer: Thank you for the helpful comment. We have clarified the description in the Methods section and Table 1 in the main text, as well as specified the dataset name as **ncAA-1** in the description. We would like to note that the definition of *therapeutic peptides* is based on their **biological and pharmacological activities**, not on the number of non-canonical amino acids (ncAAs) they contain [10].

In fact, natural peptides themselves can serve as therapeutic peptides. As discussed by Wang et al. [10], "research into therapeutic peptides started with fundamental studies of natural human hormones, including insulin, oxytocin, vasopressin, and gonadotropin-releasing hormone (GnRH), and their specific physiological activities in the human body." This illustrates that both natural and modified peptides fall under the definition of therapeutic peptides. In our study, the "ncAA-1 dataset" serves as a benchmark for evaluating structures containing ncAAs, rather than as a dedicated therapeutic peptide benchmark. Our goal is to illustrate one potential application of ncAA incorporation within the broader context of therapeutic peptide design.

Notably, several well-established examples support the relevance of single ncAA substitution in therapeutic development. For instance, in a groundbreaking study on insulin, Lieblich and Fang et al. demonstrated that a single proline-to-4S-hydroxyproline substitution accelerated monomer dissociation and reduced fibrillation compared to native insulin [11]. Clinically approved peptides such as **Liraglutide** and **Leuprolide** also contain a single modified amino acid and are widely used to treat metabolic disorders and hormone-sensitive cancers, respectively [12]. Furthermore, non-canonical residues like **norvaline** and **norleucine** have been shown to enhance peptide-protein interactions, as seen in the design of a hexapeptide binding to the PDZ domain of PSD-95 protein [13]. These cases underscore the impact that a single ncAA can have on biological function, pharmacokinetics, and therapeutic utility. We have updated the manuscript accordingly to clarify our intention and align with this perspective.

8. **Methods: The description of combinatorial Laplacian could largely be move the SI and replaced by a short summary of key point in the manuscript.**

Answer: We appreciate the reviewer's suggestion. We have revised and shortened the math method section in the previous revision. These modifications provide more intuitive explanations with concrete examples and reduce math jargon while preserving essential mathematical rigor.

In addition, we also moved a lot of basic definitions to the supporting information as suggested by the reviewer. However, we still keep the definition of the persistent combinatorial Laplacian as it plays a key role in our methodology and the interpretation of results. Including these descriptions in the main text not only supports the logical flow of the manuscript but also helps readers follow the analysis more effectively, especially for those less familiar with the math details yet genuinely interested in understanding the core of our approach. Anyway, we remain open to further moving this section to SI if the editor recommends it.

9. **Methods: It is unclear what sequences were used for AF predictions. PDB files often provide sequences but 3D coordinates are not necessarily available for all the amino acids in the sequence, both for the protein and the peptide.**

Answer: We thank the reviewer for pointing this out. As described in the Methods section (Peptide-protein complex prediction via AlphaFold2-Multimer and AlphaFold3: Protein and peptide sequences), the FASTA sequences used for AlphaFold predictions were prepared from the deposited sequences in the PDB database (i.e., full-length sequences) rather than from the crystal structure coordinates. **Both the FASTA files and crystal structures can be downloaded directly from the PDB using each PDB ID, for example, via standard Linux command-line tools such as wget.** The PDB IDs used in all datasets are available at Github Repo.

10. **Methods: It is unclear on what complexes TopoDockQ was trained and what the overlap of the training set with the validation and test sets is.**

Answer: We thank the reviewer for highlighting this question. As described in the Methods section (Dataset), TopoDockQ was trained on the SinglePPD-Training set, validated on the SinglePPD-Validation set, and tested on multiple independent datasets, including SinglePPD-Test, PFPD, LEAD-SPEP, Latest, and ncAA. The SinglePPD-Training, SinglePPD-Validation and SinglePPD-Test are split

by PDB ID from the entire SinglePPD dataset. To ensure comprehensive evaluation and avoid potential data leakage, we report in the main text results for evaluation datasets filtered at a protein–peptide sequence identity product $\leq 70\%$ against all sequences in the training set. A 70% sequence identity threshold is widely used to reduce redundancy in benchmarking datasets for both protein and peptide sequences [3–6], as it provides a practical balance between reducing redundancy and maintaining meaningful diversity, with sequence functionality shown to diverge more frequently below that level [7–9]. In the revised manuscript, we present results for **five such datasets: SinglePPD-Test_70%, PFPD_70%, LEADSPEP_70%, Latest_70%, and ncAA-1_70%**. As summarized in Table 2 of the manuscript, the resulting subsets (mean values denote the average protein–peptide sequence identity product; ranges in parentheses) comprise LEADSPEP_70% (33 complexes; mean: 0.22, range: 0.00–0.67), Latest_70% (75; mean: 0.12, range: 0.00–0.60), ncAA-1_70% (138; mean: 0.16, range: 0.00–0.70), PFPD_70% (20; mean: 0.14, range: 0.00–0.57), and SinglePPD-Test_70% (118; mean: 0.24, range: 0.00–0.68). This filtering ensures that at least one binding partner (protein or peptide) in each complex is relatively dissimilar to the training data. The SinglePPD-Validation set has a mean protein–peptide sequence identity product of 0.507 (std = 0.396) when compared to the training set. Specifically, 45.6% of the complexes have an overall identity product of $\leq 30\%$, and 60.0% have $\leq 70\%$. Detailed percentages at each 10% threshold are provided in Table 3 of the manuscript.

The datasets’ details, PDB IDs, generated structures, and features are available at Zenodo for full transparency and reproducibility. The additional sequence statistics can be found at Github Data Folder

11. Results and Discussion: Since the authors establish that DockQ is a better metric than RMSD for good docking solutions why use the ligand RMSD in the results and discussion section?

Answer: We thank the reviewer for this important question regarding our use of RMSD alongside DockQ. While DockQ is highly informative for assessing interface-level quality, RMSD remains a complementary and widely accepted metric for capturing overall structural deviations, **particularly in cases where DockQ is not applicable**. In particular, for ncAA-containing peptides, DockQ and Fnat are not defined due to the lack of standardized atom mappings, so RMSD is used as the default evaluation metric. Therefore, we rely on RMSD to evaluate structural quality in such cases. To maintain consistency, we also reported the DockQ scores of the corresponding natural peptide scaffolds from which the ncAA peptides were derived. This enables relative comparison even when direct DockQ evaluation is unavailable.

In addition, RMSD is included in the Results and Discussion section titled “AlphaFold2-Multimer and AlphaFold3 are promising for peptide–protein complex prediction” to ensure consistency with previously published benchmarking results, such as those from PIPER-FlexPepDock. This enables direct comparison with reported values in the literature and helps contextualize our results without altering or misinterpreting the original authors’ conclusions.

12. Results and Discussion: The evaluation is made on a set of 27 complexes. This is a small dataset and there apparently no check for overlap with the training set.

Answer: We thank the reviewer for raising this concern and are happy to clarify. We would like to clarify that **only one of the 5 benchmark datasets, the PFPD benchmark, contains 27 peptide–protein complexes**. The others include SinglePPD-Test (191 samples), LEADSPEP (43 samples), Latest (89 samples), and ncAA (150 samples), as summarized in Table 1 and detailed in the Methods section. Thus, the overall **evaluation was performed on a substantially larger and more diverse set of complexes**. The PFPD dataset is widely adopted and has been used in prior studies for consistent evaluation [14]. To ensure comprehensive evaluation and avoid potential data leakage, we report in the main text results for evaluation datasets filtered at a protein–peptide sequence identity product $\leq 70\%$ against all sequences in the training set. A 70% sequence identity threshold is widely used to reduce

redundancy in benchmarking datasets for both protein and peptide sequences [3–6], as it provides a practical balance between reducing redundancy and maintaining meaningful diversity, with sequence functionality shown to diverge more frequently below that level [7–9]. In the revised manuscript, we present results for **five such datasets: SinglePPD-Test_70%, PFPD_70%, LEADSPEP_70%, Latest_70%, and ncAA-1_70%**. As summarized in Table 2 of the manuscript, the resulting subsets (mean values denote the average protein–peptide sequence identity product; ranges in parentheses) comprise LEADSPEP_70% (33 complexes; mean: 0.22, range: 0.00–0.67), Latest_70% (75; mean: 0.12, range: 0.00–0.60), ncAA-1_70% (138; mean: 0.16, range: 0.00–0.70), PFPD_70% (20; mean: 0.14, range: 0.00–0.57), and SinglePPD-Test_70% (118; mean: 0.24, range: 0.00–0.68).

13. **Results and Discussion: NeuralPlexer is mentioned without any context in the middle of a paragraph about AF methods.**

Answer: We thank the reviewer for this suggestion. **NeuralPlexer was included in the context of comparing generative models for docking**, specifically in the sentence: *“In contrast, the latest diffusion model, NeuralPlexer, failed to generalize to peptide–protein complex prediction tasks under default settings on the PFPD dataset,”* we agree that additional background would be helpful in clarifying the relevance of this comparison. We have revised the paragraph to provide additional context about NeuralPlexer as a recent diffusion-based generative model trained primarily for protein–ligand complex prediction, and to explain that its failure on peptide–protein docking likely reflects this difference in design objective and training data.

14. **Results and Discussion: In the current manuscript the PPD, PFPD, LEADPEP, Latest and ncAA datasets are mentioned for the first time on page 3 in results and Discussion section without any reference, explanation of the acronym or summary of their content.**

Answer: We thank the reviewer for pointing this out. We have revised the manuscript to clearly introduce and properly reference the SinglePPD, PFPD, LEADSPEP, Latest, and ncAA datasets upon their first mention in the Results and Discussion section.

15. **Results and Discussion: Calculating both protein and ligand RMSD on AF predictions is not a RESULT but a CONSEQUENCE of AF co-folding both sequences.**

Answer: We thank the reviewer for this helpful comment and are glad to clarify our rationale. While it is true that AlphaFold-based models co-fold peptide and protein sequences, we respectfully emphasize that **RMSD is not merely a byproduct of the prediction mechanism, but a standard and informative evaluation metric in benchmarking predictive models.**

RMSD is a valid and widely used metric for assessing the deviation between predicted and experimentally determined peptide–protein complex structures, regardless of how the prediction is generated. In our study, RMSD values are calculated between predicted complexes and their reference crystal structures, and the distribution of these values offers interpretable and meaningful insights into prediction accuracy.

This approach is consistent with established practices in the field. For instance, RMSD is explicitly used to evaluate prediction quality in AlphaFold-based peptide docking studies (Tsaban et al., Nat. Commun., 2022) as well as in the AlphaFold3 framework itself (Evans et al., Nature, 2024).

16. **Results and Discussion: In the paragraph Improved Precision and Reduced False Positive Rate with p-DockQ Score: “In previous docking tests using the PFPD dataset” what does “previous docking” refer to?**

Answer: Thank you for this helpful question. The phrase “previous docking” refers to our baseline evaluation of peptide–protein complexes generated by AlphaFold2-Multimer (AF2-M) on the PFPD benchmark dataset for success rate comparison with those results reported in PIPER-FlexPepDock [14], prior to applying the proposed TopoDockQ re-ranking method. Specifically, it refers to the de-

fault ranking of predicted models based on AlphaFold’s built-in confidence metric (`af_confidence`). This comparison establishes the need for an improved scoring method, which is addressed by our p-DockQ approach.

We have revised the manuscript to clarify this point and explicitly define “previous docking” at its first mention to avoid ambiguity.

17. **Results and Discussion: “The p-DockQ score demonstrates greatly improved performance on the non-canonical peptide dataset (ncAA PEP)”. I assume this means that selecting complexes using the p-DockQ score works better than using `af_confidence`? If my understanding is correct the ncAA containing peptides were predict using natural amino acids. In this case why is it that p-DockQ performs so much better for this set than the 2 other sets? Is there a larger overlap with the training set for this dataset?**

Answer: Thank you for raising this important point. Yes, the statement refers to the improved performance of p-DockQ over the AlphaFold confidence score (`af_confidence`) in selecting near-native models for the ncAA dataset. As described in the *Methods* section, there is no overlap between the training set and any of the evaluation datasets, including the ncAA set.

The relatively better performance of p-DockQ on the ncAA dataset for some metrics reflects the generalization ability of the topological features used in TopoDockQ. While the ncAA peptides include a single non-canonical residue, the majority of the structural context remains similar to canonical peptides, enabling effective scoring for the interface. It is expected that model performance varies across different datasets due to inherent differences in their composition and complexity. To ensure robustness, we evaluated p-DockQ across five independent datasets using multiple metrics (e.g., TPR, FPR, precision), all of which show consistent improvements. It is important not to draw conclusions based solely on a single dataset or metric, as this may not accurately reflect the model’s overall performance and robustness. We discuss such into the “Improved Precision and Reduced False Positive Rate with p-DockQ Score” section under *Results and Discussion*.

18. **Results and Discussion: “accuracy remains limited because ncAA structures were not included in the training set” It is unclear why this would be the case. The approach assumes that the peptide is positioned properly while predicting the complex using canonical amino acids only. It is unclear to me that the few atoms added by the ncAA sidechain mutation are sufficient to affect the DockQ value ?**

Answer: We thank the reviewer for this thoughtful question and would be happy to clarify. **The statement refers to a possible direction for future improvement. Specifically, augmenting the training set with ncAA peptides. The reviewer’s concern may arise from a misunderstanding of this possible training set augmentation process.** When working with ncAA-containing peptides, we begin with AlphaFold-predicted peptide–protein complexes based on the natural peptide sequence. We then introduce the ncAA side chain into each of these models to generate corresponding ncAA peptide conformers. Since the initial predictions were generated without accounting for the structural effects of the ncAA, a subsequent minimization step is performed to relax the structure and resolve any steric clashes or conformational strain introduced by the non-canonical side chain. This process can lead to conformational changes, particularly at the peptide–protein interface. As a result, even a single substitution, such as replacing alanine with a bulkier ncAA, can affect the geometry and local interactions, which in turn can influence the DockQ score.

Current DockQ implementations are not applicable to complexes containing non-canonical amino acids, and we recognize this as an important limitation. As part of future work, we plan to extend the DockQ scoring framework to support ncAA-containing peptides and proteins. We have also incorporated this discussion into the “Considerations and Challenges” section under *Results and Discussion* to highlight this future direction.

19. **Results and Discussion: I could not see an improvement in the software installation instructions.**

Answer: Thank you for the helpful feedback. We have updated our GitHub Repository again at <https://github.com/XDaiNYU/TopoDockQ> and <https://github.com/XDaiNYU/ResidueX> with detailed installation instructions. We hope this resolves any earlier confusion and improves the user experience.

Reviewer 2:

All my concerns have been addressed, and the issues raised by the other two reviewers have been satisfactorily resolved. I believe the manuscript is now ready for publication.

Answer: We sincerely thank Reviewer 2 for their positive feedback and kind recommendation!

References

- [1] Tomer Tsaban, Julia K Varga, Orly Avraham, Ziv Ben-Aharon, Alisa Khramushin, and Ora Schueler-Furman. Harnessing protein folding neural networks for peptide–protein docking. *Nature communications*, 13(1):176, 2022.
- [2] Arindam Atanu Das, Om Prakash Sharma, Muthuvel Suresh Kumar, Ramadas Krishna, and Pre-mendu P Mathur. Pepbind: a comprehensive database and computational tool for analysis of protein–peptide interactions. *Genomics, proteomics & bioinformatics*, 11(4):241–246, 2013.
- [3] Kamil Kaminski, Jan Ludwiczak, Kamil Pawlicki, Vikram Alva, and Stanisław Dunin-Horkawicz. plm-blast: distant homology detection based on direct comparison of sequence representations from protein language models. *Bioinformatics*, 39(10):btad579, 2023.
- [4] Florian Gabler, Seung-Zin Nam, Sebastian Till, Milot Mirdita, Martin Steinegger, Johannes Söding, Andrei N. Lupas, and Vikram Alva. Protein sequence analysis using the mpi bioinformatics toolkit. *Current Protocols in Bioinformatics*, 72(1):e108, 2020.
- [5] Sandeep K. Dhanda, Kerrie Vaughan, Veronique Schulten, Alba Grifoni, Daniela Weiskopf, John Sidney, Bjoern Peters, and Alessandro Sette. Development of a novel clustering tool for linear peptide sequences. *Immunology*, 155(3):331–345, 2018.
- [6] Sujun Li, Haixu Tang, and Yuzhen Ye. A meta-proteogenomic approach to peptide identification incorporating assembly uncertainty and genomic variation. *Molecular & Cellular Proteomics*, 18(8 suppl 1):S183–S192, 2019.
- [7] Sarah Addou, Robert Rentzsch, David Lee, and Christine A. Orengo. Domain-based and family-specific sequence identity thresholds increase the levels of reliable protein function transfer. *Journal of Molecular Biology*, 387(2):416–430, 2009.
- [8] A. A. Aptekmann, J. Buongiorno, D. Giovannelli, M. Glamoclija, D. U Ferreira, and Y. Bromberg. mebipred: identifying metal-binding potential in protein sequence. *Bioinformatics*, 38(14):3532–3540, 2022.
- [9] Damien Devos and Alfonso Valencia. Practical limits of function prediction. *Proteins: Structure, Function, and Genetics*, 41(1):98–107, 2000.
- [10] Lei Wang, Nanxi Wang, Wenping Zhang, Xurui Cheng, Zhibin Yan, Gang Shao, Xi Wang, Rui Wang, and Caiyun Fu. Therapeutic peptides: current applications and future directions. *Signal transduction and targeted therapy*, 7(1):48, 2022.
- [11] Jia Fang, Sarah E. Lieblich, et al. Peptide engineering and design for therapeutic applications. *Current Opinion in Biotechnology*, 60:211–218, 2019.
- [12] Z. Zhou et al. Therapeutic peptides: current applications and future directions. *Signal Transduction and Targeted Therapy*, 7(1):48, 2022.
- [13] W. S. Horne et al. Structure–activity relationships and design of nonstandard amino acid peptides targeting protein–protein interactions. *Journal of Medicinal Chemistry*, 59(13):5865–5875, 2016.

- [14] Nawsad Alam, Oriel Goldstein, Bing Xia, Kathryn A Porter, Dima Kozakov, and Ora Schueler-Furman. High-resolution global peptide-protein docking using fragments-based piper-flexpepdock. *PLoS computational biology*, 13(12):e1005905, 2017.

Authors' Response to Reviews of:
Topological Deep Learning for Enhancing Peptide-Protein
Complex Prediction (Manuscript ID:
COMMSCHEM-25-0234-B)

Xuhang Dai ^{1,*}, Rui Wang* ^{2,*}, and Yingkai Zhang ^{1,2,3,†}

¹Department of Chemistry, New York University, New York, NY 10003, USA

²Simons Center for Computational Physical Chemistry, New York University, New York, NY 10003, USA

³NYU-ECNU Center for Computational Chemistry, New York University Shanghai, Shanghai, 20062, CN

Dear Editor and Reviewer:

Thank you very much for your mail of 10-Sep-2025, and referees' reports concerning our manuscript entitled: "Topological Deep Learning for Enhancing Peptide-Protein Complex Prediction" (Manuscript ID: **COMMSCHEM-25-0234B**). We have accordingly reviewed the referees' comments and addressed their concerns on package installation. In the revised manuscript, the deleted text is marked in red with a strikethrough, while added text is shown in blue with a wavy underline. Full responses to reviewers' comments are below (reviewers' comments are in bold black, while our responses are in blue).

Reviewer 1:

I thank the authors for their careful consideration of the previous critics and the modifications made to the manuscript which, as far as I am concerned, is now suitable for publication.

However, I still was unable to run the code and therefore have to take all results at face value, which is pretty disheartening given that this is now the third time this manuscript is getting to reviewers. How had can it be to go through the installation instructions and make sure they can be carried out by a potential user.

the Code Availability section provide the following link "The code of TopoDockQ feature generation is available at <https://github.com/wangru25/TopoDockQFeature>." which does not exist.

I was able to get the conda environment setup but then the 02_tutorial_inference jupyter notebook is opened (assuming people know how to do that), it prominently displays a NOTE telling users to download a file called lr.0.0005.bs_512.zip with no indication where to find it. After searching around for 20 minutes I gave up. This is rather upsetting given the repeated complaints from reviewers not being able to run the software to verify any of the results presented in the paper.

Answer: We thank the reviewer for raising the concern regarding the Code Availability. We have thoroughly re-tested the installation and provided more detailed guidance, including two YouTube videos, to assist potential readers in successfully running our code.

1. For the code of the TopoDockQ feature generation, we would like to clarify that the manuscript includes the correct link. It is possible that the reviewer may have unintentionally 1) omitted a dash between "TopoDockQ" and "Feature", and 2) added an additional period "." at the end of the link. If the reviewer clicks the hyperlink, it will directly lead to the publicly available GitHub repository. Please try again here to access the Github page: <https://github.com/wangru25/TopoDockQ-Feature>

Additionally, we have uploaded a YouTube video to guide interested readers in installing TopoDockQ and generating their own features. Please refer to https://youtu.be/fozzr7Y_bQc for the TopoDockQ feature generation video.

2. The model file previously referred to as lr.0.0005.bs_512.zip has been renamed and simplified as trained_model.zip, which contains the final trained checkpoint: best_model.pth. We have updated the GitHub repository (<https://github.com/XDaiNYU/TopoDockQ>) to clarify where users can download all necessary files:
 - Instructions for downloading the CSV data files are now provided in `data/processed_data/download_guidance.md`.
 - Instructions for downloading trained model files are available in `models/download_guidance.md`.

We also provided a YouTube video <https://youtu.be/IGOtribF4u4> for complete guidance regarding feature generation, training, and inference.

Two YouTube video links are also provided in our manuscript.